# Force and kinetics of fast and slow muscle myosin determined with a synthetic sarcomere–like nanomachine
Valentina Buonfiglio[1], Irene Pertici [1], Matteo Marcello[1], Ilaria Morotti[1], Marco Caremani [1], Massimo Reconditi [1], Marco Linari [1], Duccio Fanelli [2] ✉, Vincenzo Lombardi [1] ✉ & Pasquale Bianco [1]

Myosin II is the muscle molecular motor that works in two bipolar arrays in each thick filament of the striated (skeletal and cardiac) muscle, converting the chemical energy into steady force and shortening by cyclic ATP–driven interactions with the nearby actin filaments. Different isoforms of the myosin motor in the skeletal muscles account for the different functional requirements of the slow muscles (primarily responsible for the posture) and fast muscles (responsible for voluntary movements). To clarify the molecular basis of the differences, here the isoform–dependent mechanokinetic parameters underpinning the force of slow and fast muscles are defined with a unidimensional synthetic nanomachine powered by pure myosin isoforms from either slow or fast rabbit skeletal muscle. Data fitting with a stochastic model provides a self–consistent estimate of all the mechanokinetic properties of the motor ensemble including the motor force, the fraction of actin–attached motors and the rate of transition through the attachment–detachment cycle. The achievements in this paper set the stage for any future study on the emergent mechanokinetic properties of an ensemble of myosin molecules either engineered or purified from mutant animal models or human biopsies.

Steady force and shortening in the half-sarcomere, the functional unit of the muscle cell, are due to ATP-driven cyclic interactions of the subfragment 1 (S1, the head) of the heavy meromyosin fragment (HMM) of the myosin II molecule extending from the thick filament with the actin monomers on the nearby overlapping thin filament (Fig. 1). In each interaction the free energy of the splitting of MgATP to MgADP and inorganic phosphate (Pi) in the head is associated to a structural working stroke consisting in a tilting between the motor domain, firmly attached to actin (red), and the light chain binding domain (or lever arm, light blue) connected to the myosin filament backbone through the subfragment 2 (S2, the tail, green). In isometric contraction, lever arm tilt raises the force exerted by the half-sarcomere, increasing the strain of all the elastic elements represented for simplicity in Fig. 1, state (b) by the bending of the lever arm.

When the load is lower than the maximum steady force exerted under isometric conditions ($T_0$, that is conventionally expressed as force per cross-sectional area of the contractile material, $kN\ m^{-2}$), lever arm tilting results in relative filament sliding with a reduced strain in the elastic components (Fig. 1, state (c)). Cyclic asynchronous interactions of myosin motors with the actin filament account for the generation of steady force and shortening. The shortening velocity $V$ is inversely proportional to the force $T$ (force–velocity relation, $T - V$[1]). At physiological concentrations of ATP, ADP release is the rate-limiting step for motor detachment from actin (step (b)/(c) → (d)). The rate of ADP release is conformation-dependent, increasing during steady shortening when motors at the end of the working stroke would become negatively strained. This explains the increased rate of energy liberation $\dot{E}$ (and the underlying ATP hydrolysis rate, $\phi$) when the load is reduced and the shortening speed is increased[1–8]. Faster detachment of negatively strained (compressed) motors prevents the ones at the end of their working stroke to oppose positively strained motors, a requirement for the maximisation of the power and efficiency of an array of motors working in parallel. The performance of different types of skeletal muscles depends on the myosin II isoform expressed in the muscle. Slow muscles, which are primarily involved in maintenance of posture and are characterised by the dominant presence of the isoform 1 of Myosin Heavy Chain (MHC − 1 isoform), exhibit lower shortening speed at any given load, thus develop lower power and consume ATP at a lower rate than fast muscles, which are

[1]PhysioLab, University of Florence, Sesto Fiorentino, FI, Italy. [2]Department of Physics and Astronomy, University of Florence, Sesto Fiorentino, FI, Italy. ✉e-mail: duccio.fanelli@unifi.it; vincenzo.lombardi@unifi.it

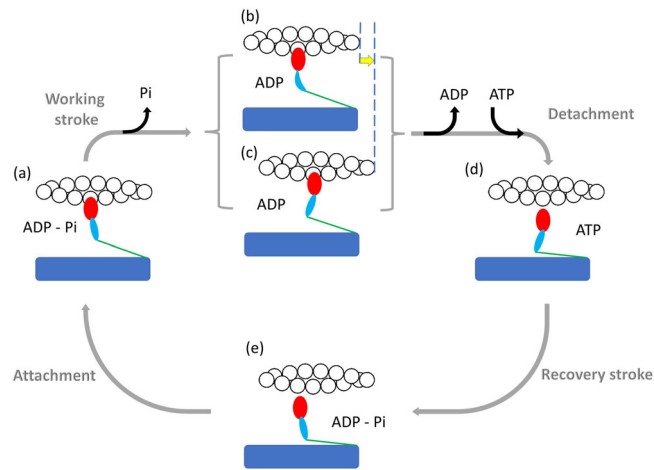

**Fig. 1 | Schematic diagram of the chemo-mechanical cycle of the myosin motor during its interaction with the actin filament.** The HMM fragment of the myosin molecule is a dimer with each monomer made by the subfragment 1 (S1 or head containing the motor domain (red) and the light chain domain (the lever arm, light blue) and the subfragment 2 (S2 or tail, green) extending from the myosin filament backbone (blue). For simplicity, only one S1 and S2 are represented here. The myosin · ADP · Pi complex attaches to actin (white circles) (**a**), forming the cross-bridge, which triggers tilting of the lever arm and Pi release with generation of force and actin filament sliding. If the mechanical load is high, it opposes filament sliding, and tilting of the lever arm causes increase in strain in the system, represented here by the distortion of the lever arm (**b**). If the load is low (**c**) tilting of the lever arm causes actin filaments sliding (yellow arrow), keeping the strain low. ADP release from and ATP binding to the motor domain cause myosin detachment from actin. ADP release is slower at high load, (**b**) → (**d**), and becomes faster at lower load (**c**) → (**d**). Hydrolysis of ATP in the detached head and reversal of the lever arm tilting (recovery stroke, (**d**) → (**e**)) complete the cross-bridge cycle. The absence of ATP causes the cycle to stop before detachment so that all motors stay attached to actin (rigor).

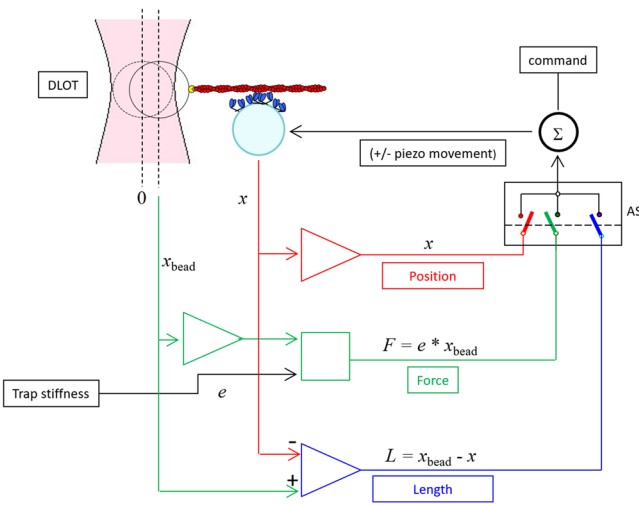

**Fig. 2 | Block diagram of the system for nanomachine mechanics.** HMM fragments (blue) deposited on the functionalised lateral surface of a pulled micropipette (cyan) are brought to interact with the actin filament (red) attached with the correct polarity ( + ) via gelsolin (yellow) to the bead trapped in the focus of the DLOT (pink). Force generation produces the movement of the bead away from the focus of the DLOT. The switch (AS) selects the feedback signal that, together with the command (black), feeds the summing amplifier Σ that drives the piezo nanopositioner: in position clamp (red) the feedback signal is the position of the nanopositioner $x$ carrying the support for the myosin array; in force clamp (green) the feedback signal is the force ($F$, calculated as the product of the stiffness of the trap ($e$) times the change in position of the bead $x_{bead}$); in length clamp (blue) the feedback signal is the change in the distance $L$ between the position of the bead and the myosin array support.

involved in movement and are characterised by the dominant presence of the $MHC - 2A, - 2B$ or $- 2X$ isoforms[9]. Strikingly, the functional difference between slow and fast isoforms is due to a difference of only 20% in the amino-acid composition.

During an isometric contraction the power is zero, so that the rate of energy consumption accounting for the steady force $T_0$ (denoted $\dot{E}_0$) corresponds to the rate of heat production ($\dot{H}_0$)[1]. $\dot{E}_0$ has been found ~fivefold larger in fast muscles than in slow muscles[10-13]. The underlying rate of ATP hydrolysis at $T_0$ can be obtained from $\dot{E}_0$ by dividing it by the energy liberated per molecule of ATP hydrolysed ($\Delta G_{ATP} = 60$ kJ mol$^{-1}$ in mammalian muscle according to[14]). In this way the energetic cost of the isometric force in the intact muscle can be compared with that in the demembranated fibres, in which the rate of energy liberation is determined by measuring the rate of ATP hydrolysis. A further normalisation for the concentration of myosin motors in the mammalian muscle (0.18 mM[11]) gives the rate of ATP hydrolysed per myosin motor ($\phi$). In demembranated fibres of rat fast muscle[15,16], rabbit[17,18] and human muscle[19] at 12 °C, $\phi$ is fivefold (or more) larger than in slow muscle, in agreement with muscle measurements (Supplementary Table 1). In both fast muscles[10-13] and fast demembranated muscle fibres[15-18] $T_0$ is either similar or at max 1.5-fold larger than in slow muscles and muscle fibres. Thus the tension cost of the isometric contraction $\dot{E}_0/T_0$ results to be systematically larger in fast muscles by on average fivefold (with a minimum of threefold). The justification for the elevated tension cost of the fast muscle can only partly be found in the intrinsic larger actin-activated myosin ATPase in solution, which for the fast myosin is twice that of the slow myosin[20].

The bulk of data characterising the energetics of slow and fast muscles at cell and tissue levels, first of all the ~fivefold larger isometric tension cost, leaves open the question of the underlying molecular mechanism. Inferring

the definition of the molecular mechanism from cell and tissue is complicated by the scaling factors related to the structural organisation of the molecular motors in the three-dimensional lattice, the co-presence of different isoforms in the same muscle and even in the same muscle fibre and the possible confounding contribution of the other sarcomeric (cytoskeleton and regulatory) proteins. Even assuming that the tension cost is solely related to intrinsic properties of the motor isoform, the question remains about the role played by the differences in the mechanokinetic properties of the motor, as the force developed in a single motor interaction or the fraction of the ATPase cycle time each motor spends attached (the duty ratio) while working in situ in the half-sarcomere of the striated muscle. In vitro, the definition of the emergent properties of the half-sarcomere, which cannot be studied with single molecule mechanics[21,22], became recently accessible by exploiting a unidimensional synthetic nanomachine powered by myosin motors purified from the skeletal muscle[23]. The nanomachine allows the performance of the half-sarcomere, the generation of steady force and shortening, to be reproduced by an ensemble of pure myosin isoforms interacting with the actin filament without the confounding effects of other sarcomeric proteins and higher hierarchical levels of organisation of the muscle. In the nanomachine, 8 HMM fragments extending from the functionalised surface of a micropipette carried on a three-way nanopositioner acting as a length transducer interact with an actin filament attached with the correct polarity to a bead trapped by a Dual Laser Optical Tweezers (DLOT) acting as a force transducer (Fig. 2). In solution with physiological ATP concentration, in which the two motors of each dimer act independently[23], myosin motors, after entering in contact with the actin filament, establish continuous interactions underpinning force development to a steady maximum value (equivalent to the force generated by the muscle in isometric contraction). In the original design[23] the system was operated either in position clamp (achieved using as feedback signal the position of the nanopositioner carrying the motor array ($x$), red branch in Fig. 2), to reproduce the isometric contraction, or in force clamp (achieved using as feedback

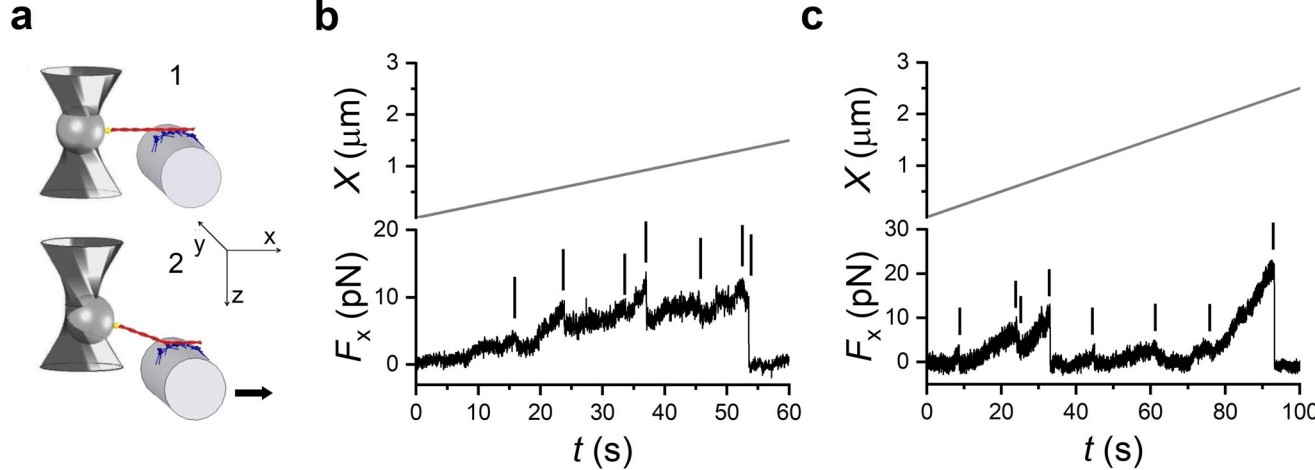

**Fig. 3 | Estimating the number of HMMs available for actin interaction from rigor rupture events in ATP-free solution. a** 1. Formation of the rigor bonds between the HMM array and the actin filament. 2. The motor support is moved away first in the direction ($z$) perpendicular to the plane of the actin–myosin interface and then in the direction ($x$) parallel to the plane, as indicated by the arrow. Panel modified from Ref. [23]. **b** Force ($F_x$, lower record) of an ensemble of soleus HMMs in

response to the movement of the nanopositioner away from the actin filament in the $x$ direction (upper record, velocity 50 nm s$^{-1}$). The small vertical bars indicate the rupture events (force drop completed in less than 50 ms), the last of which corresponds to complete detachment of the actin filament. **c** Records with the same protocol applied to an ensemble of psoas HMM.

signal the position of the bead in the laser trap ($x_{bead}$), green branch in Fig. 2), to reproduce isotonic contraction.

A major limit of the nanomachine working in position clamp was the large trap compliance in series with the motor array, two orders of magnitude larger than the native compliance in series with the half-sarcomere. This makes each addiction–subtraction of force by individual motor attachment–detachment to induce substantial sliding undermining the condition of independent force generators of the motors in the native half-sarcomere. Consequently, in position clamp, the kinetics of the attached motors is influenced by the push–pull experienced when actin slides away–toward the bead for the addition–subtraction of the force contribution by a single motor (Ref. 23, Supplementary Fig. 7).

In the present experiments the system has been implemented to operate in length clamp (achieved using as feedback signal the difference between the position of the bead and that of the nanopositioner $x_{bead} - x = L$), blue branch in Fig. 2. In length clamp the sliding between the actin filament and the motor array caused by forcegenerating interactions is eliminated because any movement of the bead is counteracted by the movement of the nanopositioner. In this way, the condition of the motors as independent force generators in the array is recovered and the rate of development of the steady isometric force, as well as the force fluctuations superimposed on the steady force, are direct expression of attachment/force-generation and detachment of the myosin motors. The data collected from either myosin isoform are used to feed a stochastic model providing a self-consistent estimate of all the relevant mechanokinetic parameters of the isometric performance of the motor ensemble: the force of a motor, $f_0$, the fraction of actin–attached motors, $r$, and the rate of transition through the attachment–detachment cycle, $\phi$, without assumptions from cell mechanics and solution kinetics as in previous studies[23–25]. The combined experimental and theoretical achievements reported in this paper set the stage for any future study on the emergent mechanokinetic properties of an ensemble of myosin molecules, either engineered or purified from mutant animal models or human biopsies.

## Results
### Estimate of the number of HMM molecules on the support available for the interaction with the actin filament
The number of motors on the micropipette surface able to interact with the actin filament ($N$) is initially determined by measuring the number of mechanical rupture events when the motor array is brought to interact with

the actin filament in ATP-free solution[23]. Following the formation of rigor bonds between the HMM-coated support and the actin filament (panel 1 in Fig. 3a), the HMM support is moved away from the actin filament, first by 1–2 $\mu$m in the direction orthogonal to the plane of the support, in order to raise a force from the trapped bead to the first bound HMM at an angle greater than 30° with the plane of the support, and then in the direction parallel to the plane, at constant velocity (50 nm s$^{-1}$), to pull the motors away from the actin filament diagonally. This allows the first bonded HMM to undergo a pulling force that is higher than the axial component shared among the other motors. In this way the myosin–actin bonds break one at a time and the attached motors cannot bind back once detached. Moreover, following each detachment the force drops because the length of actin filament segment between the bead and the next attached motor is transiently increased. Thus an additional pull is necessary to get to the next rupture event, the occurrence of which will vary in time according to the distance between the two neighbouring motors. With HMMs purified from soleus muscle, the number of rupture events per interaction (Fig. 3b) attains a saturating value of 7.9 ± 1.1 ($n = 8$), with [HMM] used to coat the pipette of 0.2 mg mL$^{-1}$. A similar saturating value of rupture events, 8.1 ± 1.4 ($n = 8$), is obtained for the HMM purified from psoas muscle with a [HMM] of 0.1 mg mL$^{-1}$ (Fig. 3c). Notably, similar saturating values of [HMM] and number of rupture events (8.1 ± 1.2) were found for the psoas motors in the previous study in which an optical fibre etched to the same diameter was used as support[23]. In 2 mM [ATP] each head of an HMM dimer works independently and thus the number of available motors is twice the number of HMM ruptures: $N = 16 \pm 2$ and $16 \pm 3$ for the soleus and psoas, respectively.

### Isometric force development by the nanomachine powered by slow and fast myosin motors
The experiment starts in position clamp, because, for the system to operate in length clamp, it is necessary that first the feedback loop is closed by the establishment of actin–myosin interactions. When an array of motors from the slow soleus muscle is brought to interact with a bead-tailed actin filament in solution with 2 mM ATP (Fig. 4a), the establishment of continuous ATP-driven actin–myosin interactions causes the force ($F$, blue trace) to rise pulling on the actin filament, which in position clamp (HMM support position $x = 0$, red trace), slides in the shortening direction ($\Delta L$, black trace, negative for shortening) due to the trap compliance (phase 1). A steady maximum force ($F_0$) of ~12 pN is attained with a shortening of ~55 nm. The

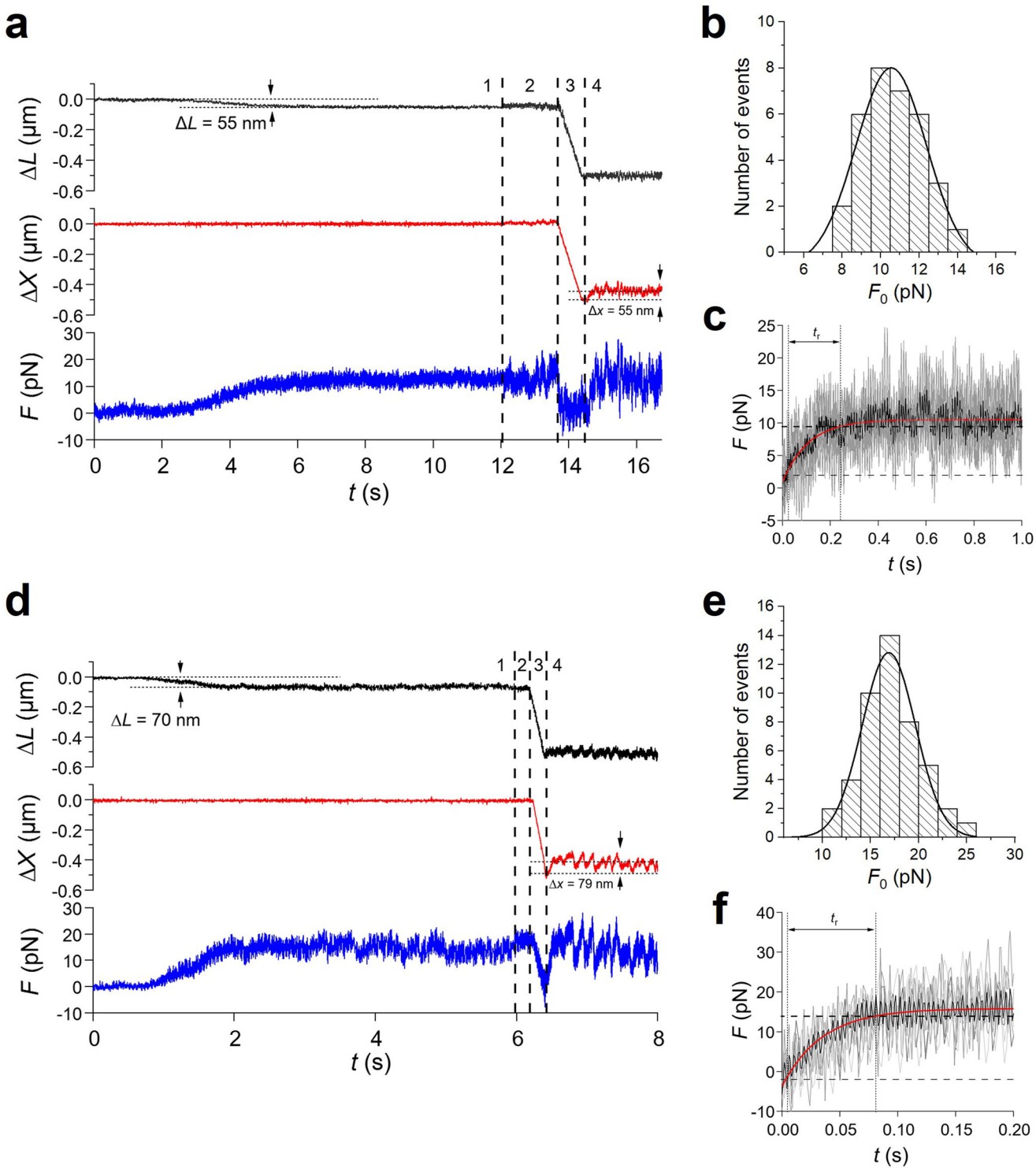

control is switched to length clamp in correspondence of the vertical dashed line separating phase 1 and 2. The switch time is marked by the increase in noise of the force trace as a consequence of the reduction of the compliance in series with the motor system. In fact, in length clamp the force change generated in each individual attachment and detachment is no longer dissipated in filament sliding against the large in series trap compliance. A shortening of ~500 nm completed within ~700 ms is superimposed on the steady isometric force in correspondence of the second vertical dashed line to drop and keep the force at zero (phase 3). When actin filament sliding

stops (third vertical dashed line) force starts to redevelop towards $F_0$ (phase 4) with just a minimum delay, indicating that the motor array was able to cope with the imposed 500 nm shortening maintaining continuous interactions under zero load. The extent of shortening minus the amount taken by the trap compliance, $(500 - 55 =)$ 445 nm, divided by the time passed from the imposition of the shortening to the start of force redevelopment (0.88 s) gives an estimate of the velocity of unloaded shortening ($V_0$) of 0.5 $\mu m\ s^{-1}$. Force redevelopment in length clamp is much faster than the original force rise in position clamp and occurs in truly isometric

**Fig. 4 | Active force generation by the nanomachine powered by slow (soleus) and fast (psoas) myosin motors. a–c** Slow myosin array. **a** Force (*F*, blue trace), movement of the nanopositioner carrying the motor array (Δ*x*, red trace) and relative sliding between the motor array and the actin filament (Δ*L*, black trace) during the actin myosin interaction. Phase 1: following the establishment of the contact between the actin filament and myosin motors, the force rises in position clamp to the maximum isometric value ($F_0 \simeq 12$ pN), with the simultaneous sliding of the actin filament by ~55 nm toward the shortening direction to load the trap compliance. Phase 2: the switch to length clamp (marked by the first vertical line) is followed by the increase in force fluctuations superimposed on $F_0$. Phase 3: force drops to zero in response to a rapid shortening of ~500 nm imposed in length clamp (start marked by the second vertical line) with actin filament sliding under zero force. Phase 4: following the end of the imposed shortening (marked by the third vertical line) force redevelops in length clamp with the nanopositioner moving by ~55 nm to counteract the trap compliance and keep the filament sliding at zero. **b** Frequency distribution of $F_0$. Data are plotted in classes of 1 pN and fitted with a Gaussian (continuous line) with centre 10.5 pN and standard deviation $\sigma = 1.8$ pN. **c** Time course of force redevelopment after rapid shortening (black trace) averaged from 6 records from as many experiments (grey traces). The red line is the single exponential fit to measure $t_r$ (the time elapsed from 10%, horizontal thin dashed line, to 90%, thick horizontal dashed line, of $F_0$ recovery). **d–f** Fast myosin array. **d** *F*, Δ*x* and Δ*L*, defined and colour coded as in (**a**). Phases 1–4 as described in (**a**). **e** Frequency distribution of $F_0$ plotted in classes of 2 pN and fitted with a Gaussian (continuous line) with centre 17 pN and standard deviation $\sigma = 3$ pN. **f** Time course of force redevelopment after rapid shortening (black trace) averaged from 7 records from as many experiments (grey traces). The red line is the single exponential fit to measure $t_r$ labelled as in (**c**).

conditions, as the movement of the bead due to trap compliance is counteracted by a corresponding movement of the nanopositioner in the lengthening direction (red trace, ~ 55 nm), that keeps Δ*L* = 0 (black trace). Notably, the force redevelopment following a 500 nm release attains the same $F_0$ value as that attained during the original rise in position clamp thanks to the architecture of the machine, in which the dimension of the motor array remains constant independent of the amount of reciprocal sliding[23]. $F_0$ from 33 records shows a Gaussian distribution with centre 10.5 pN (Fig. 4b). The rate of force redevelopment in length clamp only depends on the attachment/detachment kinetics of myosin motors in isometric conditions. Force redevelopment is roughly exponential, and its time course is quantified by the rise time $t_r$ (the time from 10% to 90% of $F_0$). $t_r$ estimated on the record (Fig. 4c, black) obtained by averaging 6 traces from as many experiments (grey) is 238 ± 13 ms. The time constant of the underlying exponential force rise of the soleus-powered nanomachine, $\tau$, is ($t_r/2.2 =$) 108 ± 5 ms, and the rate of force development, *a* is (1/$\tau$ =) 9.3 ± 0.5 s$^{-1}$. The sequence of events accompanying the interaction of the array of motors purified from psoas muscle with the actin filament is the same as for the soleus motors (Fig. 4d). The force develops in position clamp (phase 1), while the actin filament slides in the shortening direction due to the trap compliance. A steady isometric force $F_0$ (15.9 pN), is attained with a shortening of 70 nm. In the 47 records of the psoas HMM, $F_0$ shows a Gaussian distribution with centre 17 pN (Fig. 4e). Following the switch to length clamp, a rapid shortening of ~ 500 nm is imposed so that the force drops to zero. The shortening in this case is just sufficient to drop the force to zero, given the much faster shortening velocity afforded by the fast motor array, so that, as soon as the actin filament sliding stops (third vertical dashed line), $V_0$, calculated by the extent of shortening minus the amount taken by the trap compliance, (500 − 70 =) 430 nm, divided by the time passed from the imposition of shortening to the start of force redevelopment (0.22 s), is 1.95 µm s$^{-1}$ (3.9 times larger than that of slow muscle). It must be considered, however, that $V_0$ in this case is somewhat underestimated, as most of the shortening occurs with force greater than zero. Force redevelopment in length clamp (phase 4) occurs with a rate that is not influenced by the trap compliance and thus is the expression of the kinetics responsible for the transition to the steady force $F_0$ by the fast isoform array. A $t_r$ of 77 ± 4 ms is estimated on the record (black in Fig. 4f) obtained by averaging the traces from 7 experiments (grey). The corresponding $\tau$ (=$t_r/2.2$) and *a* (=1/$\tau$) are 35.0 ± 1.8 ms and 28.6 ± 1.4 s$^{-1}$, respectively.

The upper - 3 dB frequency characterising the force rise $f_c = 0.35/t_r$ is 4.5 ± 0.2 Hz.

Two main points emerge from these measurements on the synthetic machine operating in length clamp conditions. The first is that the rate of force redevelopment, which only depends on the attachment/detachment kinetics of myosin motors in isometric conditions, is three times slower in the soleus powered nanomachine than in the psoas powered nanomachine. The second point is that the force fluctuations around the average value displayed by the force record at the steady state are stemming from individual attachment/detachment events. Both pieces of information can be used to feed the stochastic model described in the next section.

## Modelling the mechanical output of the nanomachine powered by the slow and the fast isoform ensembles

In the stochastic model each motor exists in three possible states (or motor configurations): one detached state and two different force-generating attached states. Fitting the experimental records allows a self-consistent estimate of all the relevant mechanokinetic parameters of the nanomachine including the force exerted by a single myosin motor and the average number of attached motors in the stationary state, without assumptions from cell and solution kinetic studies.

As detailed in the Introduction, the implementation of the length clamp mode allows us to recover the condition of the motors as independent force generators in the array. Therefore we consider an ensemble of *N* independent ATP-fuelled molecular motors interacting with an actin filament in isometric conditions. Each motor can be found in the detached state *D*, in the attached low force-generating state $A_1$, or in the attached high force-generating state $A_2$. The corresponding kinetic scheme, which exemplifies the possible transitions between distinct allowed motor states is:

$$D \underset{k_{-1}}{\overset{k_1}{\rightleftharpoons}} A_1 \underset{k_{-2}}{\overset{k_2}{\rightleftharpoons}} A_2 \overset{k_3}{\rightarrow} D \tag{1}$$

The rate constants $k_j$, $j \in \{1, -1, 2, -2, 3\}$ represent the probability per unit of time for the reaction *j* to occur, and are expressed in s$^{-1}$. The state of the system at any time *t* is characterised by the vector $\boldsymbol{n}(t) = (n_D(t), n_1(t), n_2(t))$ whose entries specify the number of molecular motors in each of the considered configurations. Specifically, $n_D$ stands for the number of motors in the state *D*, $n_1$ is the number of motors in the state $A_1$ and $n_2$ denotes the number of motors in the state $A_2$. The system admits the obvious conservation law $N = n_D + n_1 + n_2$ where *N* stands for the total number of motors in any of the considered states. Accounting for the above relation enables one to employ just two scalar (discrete) entries to photograph the state of the system, namely $\boldsymbol{n}(t) = (n_1(t), n_2(t))$. Under the Markov hypothesis, the stochastic dynamics of the scrutinised system is ruled by a master equation which sets the evolution of the probability $P(\boldsymbol{n}, t)$ of finding the system in the state specified by the vector $\boldsymbol{n}$ at time *t*. The master equation accounts for the balance of opposing contributions: on the one side the transitions *towards* the reference state (the associated terms bearing a plus sign). On the other, the transitions *from* the reference state (terms with a minus). From the master equation one can readily derive the mean field equations that govern the deterministic dynamics for the continuous concentrations *y* and *z* of the molecular motors in states $A_1$ and $A_2$, as detailed in Methods. Further, a self-consistent elimination of the variable *y* can be performed to yield a simpler description of the examined process in terms of the variable *z* (the derivation is given in Methods, see also Supplementary Fig. 1):

$$z(t) = \frac{b}{a}\left(1 - e^{-at}\right) = \frac{k_1}{k_1 + G} \frac{k_2}{k_2 + k_{-2} + k_3}\left(1 - e^{-\left[k_{-2} + k_3 + \frac{k_2(k_1 - k_{-2})}{k_1 + k_{-1} + k_2}\right]t}\right) \tag{2}$$

where $a$ and $b$ are positive quantities, self-consistently defined by the latter equality and $G = (k_{-1}(k_{-2} + k_3) + k_2 k_3)/(k_2 + k_{-2} + k_3)$. We define the duty ratio $r$ as the fraction of attached motors (or the fraction of the ATPase cycle time a motor spends attached). Assuming that for mammalian muscle myosin at temperature $T \simeq 24$ °C motors are prevalently found in the state $A_2$, a straightforward analysis outlined in the Methods yields $r \simeq z^* = \frac{k_1}{k_1 + G}$. Here, $z^*$ stands for the equilibrium concentration of $A_2$ motors. Through parameter $a$, we have also access to a closed estimate for the characteristic time scale of the exponential evolution of $z$. Let us notice that $a$ is the inverse of the time constant of the development of the steady force, $\tau$ as defined in the experiment, hence $a = 2.2/t_r$. According to this simplified scheme, the effective rate of ATP consumption can be estimated as the flux of motors through the cycle per unit time. This equals to the rate of motors in $A_2$ detaching from the actin, in formula $\phi = z^*(a - b)$.

Starting from these premises we can characterise the average force exerted by a small ensemble of molecular motors in isometric conditions. This is obtained by combining the contributions from each individual motor of the collection: motors in the configuration $A_1$, each exerting a force $f_1$ and motors in $A_2$, each exerting a force $f_2$. In the nanomachine, motors have a random orientation with respect to the actin filament. As a consequence, we assume that the intensity of the force exerted by a motor depends on the binding angle $\theta$, as measured from the correct orientation. Depending on the specific orientation of the molecule, the force progressively decreases up to a minimum value that is $0.1 f_0$[22]. In particular, the force of a single motor can change within a bounded interval: the largest value of the force $f_0$ is exerted when the motor orientation is correct (corresponding to the in situ orientation). Then, in accordance with[24] (see Supplementary Fig. 2) we postulate that the exerted force $f_1$ is a random variable, uniformly distributed within the interval $\mathcal{I}_1 = [-f_0, f_0]$. Similarly, the force $f_2$ is randomly extracted from the interval $\mathcal{I}_2 = [\frac{f_0}{10}, f_0]$. The mean field average force exerted by the ensemble of myosin motors, at any time $t$, is $\langle F(t) \rangle = \langle n_1(t) \rangle \langle f_1 \rangle + \langle n_2(t) \rangle \langle f_2 \rangle = \langle n_2(t) \rangle \langle f_2 \rangle$, given that $\langle f_1 \rangle = 0$ since the interval $\mathcal{I}_1$ is symmetric with respect to zero. In the stationary state, $\langle F(t) \rangle$ converges to the asymptotic plateau $F_0$, and thus:

$$r f_0 = \frac{1}{N} \frac{20}{11} F_0$$

where use has been made of the conditions $\langle f_2 \rangle = (11/20) f_0$ and $z^* \simeq r$.

The experimental value of the stationary force exerted by a pool of $N$ motors acting in the state $A_2$ solely constrains the product of $f_0$ and $r$. That is, on deterministic means, we cannot access a direct estimate of the maximum force exerted by an individual motor ($f_0$) and the associated duty ratio ($r$), but just constrain this latter pair to fall on a hyperbole, set by $F_0$. Accounting for the fluctuations superimposed on $F_0$, and thus by properly gauging the stochastic component of the dynamics, enables us to resolve the above degeneracy.

To this end we consider the dynamics of the system at finite $N$, so as to account for the role played by finite size fluctuations. To quantify the contribution as stemming from the intimate graininess of the investigated system, we ought to solve the master equation, focusing in particular on the stationary state probability distribution $\boldsymbol{P}^{st}$. As discussed in Methods, we are in a position to solve exactly the stochastic model in its stationary state, and thus get a closed expression for $\boldsymbol{P}^{st}$, as function of the parameters of the model. This knowledge can be used to compute $P(F)$ the distribution of the exerted force $F$ (see Methods). We remark that $P(F)$ is ultimately shaped by the kinetic constants of the model (namely, $k_1, k_{-1}, k_2, k_{-2}, k_3$) and also reflects the maximum force $f_0$, as applied by individual motors. Recall that $N$ is directly determined (Fig. 3). We can therefore construct an inverse procedure to recover information on the underlying parameters, by confronting the predicted distribution of the force $P(F)$ to the homologous curve experimentally recorded. In particular we will prove that, by exploiting the information stemming from the fluctuations, it is eventually possible

**Table 1 | Average values of the relevant parameters estimated by the stochastic model**

| Estimated Parameters | fast | slow | ratio |
|---|---|---|---|
| $f_0$ (pN) | 6.8 ± 1.0 | 2.4 ± 0.4 | 2.8 |
| $r$ | 0.32 ± 0.02 | 0.50 ± 0.03 | 0.64 |
| $\phi$ (s$^{-1}$) | 6.0 ± 0.2 | 2.27 ± 0.04 | 2.6 |

Average values (mean ± SD obtained by averaging over 6 data records for each isoform) of the three relevant parameters estimated by the stochastic model: the force of a motor, $f_0$, the fraction of actin–attached motors, $r$, and the rate of transition through the attachment–detachment cycle, $\phi$.

to unambiguously determine both $f_0$ and $r$ (see Supplementary Table 2, Supplementary Figs. 8 and 9). The relevant steps that define the envisaged fitting strategy are summarised in the following and made explicit in the Methods:

1. The first step amounts to analyse the time evolution of the force in its mean field approximation: the asymptotic force $F_0$ and the time scale $a$, as defined above, are extracted via a direct – two parameters – fit that exploits expression (2).
2. We turn to study the distribution of the fluctuation of the force around the equilibrium value. To this end we make use of $\boldsymbol{P}^{st}$.
3. From $\boldsymbol{P}^{st}$ we extract the $N + 1$ marginal probabilities $\rho_k$, namely the probabilities to find $k \leq N$ motors in the force-generating configuration $A_2$. This is achieved by summing over $n_1 = 0, \ldots, N$ the stationary probability distribution $\boldsymbol{P}^{st}$.
4. We then make use of the marginal probabilities ($\rho_0, \rho_1, \rho_2, \ldots, \rho_N$) to weight the probability distributions $\Pi_k(f)$ of the force exerted by a set of $k$ motors. These latter are computed as generalised Irwing-Hall distributions for independent and identically distributed random variables $f$ drawn from the considered interval $\mathcal{I}_2$. The distribution of the force is hence estimated as $P(F) = \sum_{k=0}^{N} \rho_k \Pi_k(f)$.
5. For fixed size $N$ (previously estimated by the counting of rupture events in ATP-free solution, see also Supplementary Fig. 10 where the possibility to modulate $N$ is accounted for) we adjust the kinetic rate constants $k_1, k_{-1}, k_2, k_{-2}, k_3$, so as to minimise the root mean square distance between the recorded distribution and its analytic estimate. The best fit values are used to compute the parameter $b$ and thus determine the sought estimates for $r$ and $f_0$, as well as the rate of motors detaching from the actin, i.e. $\phi = z^*(a - b)$.

The above fitting strategy is successfully challenged against synthetic data as discussed in Methods. Then we proceed by applying the validated procedure to the experimental data collected with the nanomachine powered by either of the HMM isoforms. As mentioned, the number of available molecular motors ($N = 16$) estimated from number of ruptures in rigor for both isoforms (Fig. 3), is assumed as the reference value in the following, unless otherwise specified. We interpolate the distribution of the fluctuations as experimentally recorded, given the analytical solution obtained above. A representative example of the fitting outcome for the soleus HMM ensemble is reported in Supplementary Fig. 11. The estimated values for $f_0$, $r$ and $\phi$, for both the psoas and the soleus HMM, are listed in Supplementary Note 1 (Supplementary Tables 3 and 4, respectively), and their respective average values are reported in Table 1. In Fig. 5 the results of the analysis are plotted in the parameters plane ($f_0$, $r$) (symbols and lines refer to different isoforms according to the colour: blue for psoas, red for soleus; different tones identify different experiments). The solid lines highlight the ensemble of distinct – though equivalent – solutions ensuing from the average force profile. By accounting for the fluctuations one breaks the degeneracy inherent to the system when analysed in its mean field version, getting just one pair ($f_0$, $r$) (identified by the symbol) compatible with each individual experimental curve.

One can relax the constraint $N = 16$ obtained from the rigor experiments (Fig. 3) and scan the range of $N$ that yields convergence of the

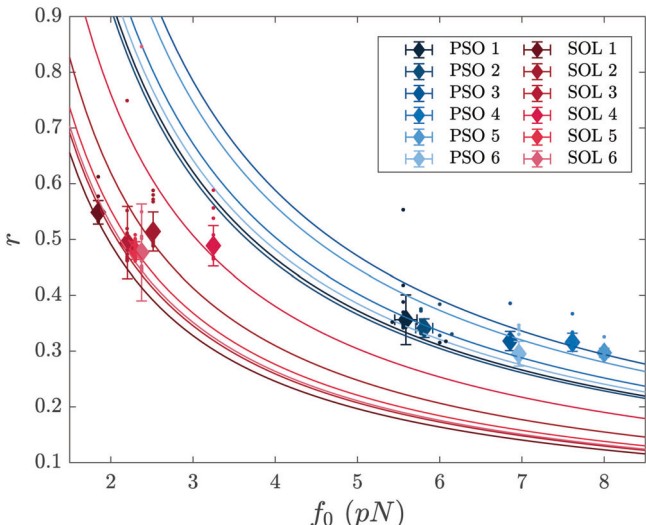

**Fig. 5 | Estimated motor force $f_0$ and fraction of attached motors $r$ from the experimental data.** Best fit parameters from the experimental data sets of rabbit soleus HMMs (red symbols) and rabbit psoas HMMs (blue symbols). Mean values and standard deviations are obtained by averaging over 20 independent realisations of the stochastic fitting procedure for each data record (coloured small dots). Different tones refer to different experiments. Each solid line represents the hyperbola on which each of the pair $(f_0, r)$ is constrained to be, according to the mean field analysis.

optimisation algorithm, for the imposed level of accuracy. The results of the analysis for the soleus isoform is reported in Supplementary Note 1 (Supplementary Fig. 12), where the estimated $f_0$ is plotted against $N$. The histogram computed from the collection of fitted parameters can be conceptualised as an indirect imprint of the degree of experimental variability as associated to $f_0$ and $N$.

## Discussion

We use the one-dimensional synthetic nanomachine described in Ref. 23 to define the isometric mechanical output of an array of 16 myosin motors purified from either fast (psoas), or slow (soleus), muscle of the rabbit. To eliminate the large trap compliance and recover the condition for the motors to operate as independent force generators as in the native half-sarcomere, once the interaction is established the system control is switched from position clamp to length clamp. The array of 16 motors in physiological ATP concentration (2 mM) at 24 °C exhibits a steady isometric force that in the fast isoform is 17 pN, and in the slow isoform is 10.5 pN. The finding that the force exerted by the same number of motors is 1.6-fold larger in the fast isoform disagrees with the most common finding in muscles and muscle fibres that the isometric force normalised for the cross sectional area of the fibre $T_0$ is either similar or at max 1.5-fold larger in the fast isoform[10–13,15–19]. Notably, in skinned fibres from the same rabbit muscles from which the nanomachine motor proteins are purified, $T_0$ in psoas at 25 °C has been found 317 ± 14 kPa, 1.9-fold larger than $T_0$ in soleus, 165 ± 12 kPa[26].

Recording the development of the steady isometric force in length clamp eliminates the contamination of the large trap compliance, showing a roughly exponential time course characterised by the parameter $t_r$ that is 238 ms for the slow isoform and 77 ms for the fast isoform. Thus the rise of the force to the maximum steady value takes a threefold longer time for the slow isoform than for the fast isoform. How this emergent property of the motor ensemble relates to the corresponding event in situ and how it is affected by the different isoforms has been tested by comparing the nano-machine output with that of $Ca^{2+}$-activated skinned fibres, from the same rabbit muscles from which the motor proteins were purified. According to

the sarcomere-level mechanics for skinned fibres developed in our laboratory, the compliance of the attachments of the skinned fibre segment to the transducer levers is negligible (see Supplementary Note 2, Supplementary Fig. 13). Under these conditions, the force redevelopment following a fast shortening able to drop the isometric force to zero is characterised by a $t_r = 265 ± 15$ ms in soleus fibres and $62 ± 5$ ms in psoas fibres. Thus, the time course of force development recorded by the nanomachine in length clamp and its modulation by the two isoforms, are in quite satisfactory agreement with those recorded at the cell level. The corresponding rates of force development ($a$) by the nanomachine are $28.6 s^{-1}$ for the fast isoform and $9.3 s^{-1}$ for the slow isoform. Considering that in length clamp $a$ is direct expression of the sum of the effective rate constant of attachment/force-generation and the effective rate constant of detachment of the myosin motors, we conclude that the interaction kinetics in isometric condition is threefold higher in the fast isoform than in the slow isoform. The attachment/detachment kinetics is expected to increase if the load on the motor ensemble is reduced, due to the strain-dependent increase in rate constant of detachment, which underpins the maximum velocity of shortening ($V_0$) attained under zero load. $V_0$ estimated by the time taken by the ensemble to redevelop force following a release able to drop the isometric force to zero (Fig. 4a, d) is $0.5 \mu m\,s^{-1}$ and $1.95 \mu m\,s^{-1}$ in the slow and fast HMM respectively, showing a $V_0 \sim 4$-fold larger in the fast isoform. Thus, the isoform-dependent increase of $V_0$ is 33% larger than the increase in $a$ and even larger if one considers that $V_0$ of the fast isoform is underestimated by the proportionally larger fraction of time spent for the force to drop to zero following the release (compare the records in Fig. 4a, d). This suggests that the fast isoform exhibits a specifically larger strain dependence of the detachment rate constant.

The rate of development of the isometric steady force and the force fluctuations superimposed on the steady force in length clamp have been exploited to implement a stochastic three-state model which is able to fit the experimental responses, allowing self-consistent estimates of all the relevant mechanokinetic parameters underlying the isometric performance of the motor ensemble: $f_0$, the force of a single correctly oriented motor, $r$, the fraction of attached motors, and $\phi$, the rate of transition through the attachment/detachment cycle (Table 1 in the Results). $f_0$ of the fast isoform ($6.8 ± 1.0$ pN) is 2.8-fold larger than $f_0$ of the slow isoform ($2.4 ± 0.4$ pN), while the ensemble force $F_0$ is only 1.6 times larger (Fig. 4b, e). This is in a great part explained by the different fraction of attached motors $r$, which in the fast isoform ($0.32 ± 0.02$) is 0.64 that of the slow isoform ($0.50 ± 0.03$). The corresponding number of attached motors ($Nr$) is $\sim 5$ and $\sim 8$ for the fast and the slow isoform respectively. The average force of a single randomly oriented motor ($0.55 f_0$) is 3.7 pN for the fast isoform and 1.3 pN for the slow isoform, from which the predicted ensemble force is ($3.7 \times 5 =$ ) 18.5 pN and ($1.3 \times 8 =$ ) 10.4 pN, respectively. These values are in quite good agreement with the observed values: $17 ± 3$ pN ($\sigma$) for the fast isoform and $10.5 ± 1.8$ pN ($\sigma$) for the slow isoform.

The model predicts a rate of transition of a motor through the interaction cycle, and thus a frequency of ATP splitting per motor ($\phi$) 2.6 times higher for the fast isoform ($6.0 s^{-1}$) than for the slow isoform ($2.3 s^{-1}$) (Table 1). The value of $\phi$ of the slow isoform array is in a remarkably good agreement with that estimated on the slow muscle of mouse and rat ($2.3$–$2.9 s^{-1}$, Supplementary Table 1). On the other hand, $\phi$ for the fast isoform array is less than half of the one estimated in the fast muscle of the same animals ($12.4$–$13.3 s^{-1}$, Supplementary Table 1). The same discrepancy for the isoform-dependent increase in $\phi$ is found between the model prediction and the skinned fibre experiments (Supplementary Table 1, Refs. [15–19]). However, it must be noted that ($i$) the absolute values of $\phi$ in skinned fibres is 10–fold smaller than the one in the muscle for both slow and fast myosin isoforms[15–18]; ($ii$) the difference can only in minor part be explained by the different temperature of the experiments (21–27 °C for the muscle and 12 °C for the skinned fibres), taking into account that the $Q_{10}$ of $\phi$ is ≤2.5 in either preparation[11,19,27]. $\phi$ predicted by the model for the output of the fast isoform nanomachine is 2.5-fold larger than that predicted for the

slow isoform, but still twofold smaller than that indicated by the energy rate measured in the fast muscle. Thus the fivefold larger $\phi$ of the fast isoform with respect to the slow isoform found in muscle is only partly explained by higher rate constants of transitions through the conventional attachment/ force generation and detachment cycle operating in isometric conditions and recorded by the nanomachine force fluctuations. The actin–activated myosin ATPase activity in solution is 2.5 times larger in fast than in slow muscle[20], which can be accounted for by a higher rate of ADP release (which is followed by a fast ATP binding and detachment, step (c)–(d) in Fig. 1[28]) and/or a higher rate of the hydrolysis step (d)–(e), and/or a higher rate of actin attachment (step (a)–(b)). In isometric contraction at physiological ATP concentration, ADP release is the rate-limiting step for detachment and is 10-fold slower in slow myosin than in fast myosin[28] and this may per se explain the finding that during steady isometric force generation the duty ratio of the fast myosin nanomachine is lower than that of the slow myosin nanomachine. However, it must be taken into account that under isometric conditions (or high load) the transitions through the different force-generating states of the motor (step (a)–(b)/(c) in Fig. 1[29,30]) slow down due to the strain dependence of the transition rate and thus the subsequent conformation-dependent release of ADP also gets slower ([5,31]). As far as the difference in $\phi$ between slow and fast myosin ensembles in isometric contraction, the finding that the force of fast myosin is 2.5-fold higher should suggest that the equilibrium distribution between different force-generating states is shifted toward the end of the working stroke in the fast myosin, in this way explaining a larger flux through the detachment step and thus the reduction in the duty ratio and the increase in $\phi$ with respect to the slow myosin (Table 1). However, it must be considered that the stiffness of the myosin motor, determined in situ with fast sarcomere-level mechanics applied to skinned fibres from rabbit muscle, is larger in the fast muscle in proportion to the motor force, so that the extent of the force-generating structural change is the same in either fast or slow myosin motor[32].

In conclusion, the 2.5-fold larger isometric $\phi$ of the fast myosin isoform found with the analysis of force fluctuations is accounted for by an intrinsic faster rate of the relevant kinetic steps of the fast myosin isoform which underpins a 2.5-fold larger ATPase rate in solution[20]. Instead, the fivefold larger isometric $\phi$ of the fast isoform reported in the literature (Supplementary Table 1), exceeds by a factor of 2 the one recorded by the nanomachine force fluctuations at 24 °C and could be explained by a further kinetic adaptation of fast myosin isoform hypothesising that, also in isometric conditions, a futile faster actin-activated ATPase cycle is present. In terms of the kinetic scheme in Ref. 33, this cycle implies the working stroke transition to occur in the motor undergoing weak actin–binding interactions and does not imply strong/force-generating attachment unless the load is reduced and the muscle shortens.

A comparison of the parameters estimated in this work with those obtained in previous nanomechanical approaches is possible for the fast isoform purified from rabbit psoas investigated by Yanagida's group[22] through the microneedle manipulation technique. In close-to-isometric conditions, obtained through a stiff microneedle, both the force of the motor (5.9 pN) and the fraction of actin-attached motors (0.36) estimated in that work are in exceptional good agreement with the values calculated here from the output of the nanomachine. A peculiar difference that makes our nanomachine unique is the possibility to define the performances emerging from the array arrangement of the motors in the half-sarcomere, as the force–velocity relation and the maximum power output. The novelty of the present nanomachine application in relation to the previous ones[23–25], is the interpretation of the output of the motor ensemble and of the isoform-dependent differences on the basis of the mechanokinetic molecular properties of either isoform in a self-consistent way without any assumptions from cell mechanics and solution kinetics. The combined experimental and theoretical achievements in this paper set the stage for any future studies on the emergent mechanokinetic properties of the half-sarcomere-like arrangement of any myosin motors, either engineered or purified from mutant animal models or human biopsies.

## Methods

### Preparation of proteins

Adult male rabbits (New Zealand white strain), provided by Envigo, were housed at Centro di servizi per la Stabulazione Animali da Laboratorio (CeSAL, University of Florence), under controlled conditions of temperature (20±1) °C and humidity (55 ± 10)%, and were euthanized by injection of an overdose of sodium pentobarbitone (150 mg kg$^{-1}$) in the marginal ear vein, in accordance with the Italian regulation on animal experimentation (Authorisation 956/2015-PR) in compliance with Decreto Legislativo 26/ 2014 and EU directive 2010/63. Three rabbits were used for the experiments. HMM fragments of myosin were purified from rabbit soleus and psoas muscles as reported previously in[23,24]. The functionality of the purified motors was always preliminarily checked with in vitro motility assay. Actin was prepared from leg muscles of the rabbits according to[34], and poly-merised F-actin was fluorescently labelled by incubating it overnight at 4 °C with an excess of phalloidin-tetramethyl rhodamine isothiocyanate[35]. For the mechanical measurements in the nanomachine, the correct polarity of the actin filament was pursued by attaching the + end of the filament to a polystyrene bead (3 $\mu m$ diameter) (Bead-Tailed Actin, BTA,[36]) with either the $Ca^{2+}$-sensitive capping protein gelsolin[23] or the $Ca^{2+}$ insensitive gelsolin fragment $TL40$ (Hypermol, Germany)[24,25].

### Mechanical experiments

The mechanical apparatus, described in detail in[23], is depicted in Fig. 2. HMM fragments of myosin were deposited randomly on the lateral surface of a glass pipette pulled to a final diameter of ~ 3–4 $\mu m$ and functionalised with nitrocellulose 1% (w/v). The glass pipette was mounted in the flow chamber carried on a three-way piezoelectric nanopositioner (nano-PDQ375, Mad City Lab, Madison WI, USA) that acts as a displacement transducer, and was brought to interact with a BTA trapped in the focus of a Dual Laser Optical Tweezers (DLOT) that acts as a force transducer. The DLOT system has a dynamic range for both force (0−200 pN, resolution 0.3 pN) and displacement (0−75 $\mu m$, resolution 1.1 nm) adequate for the measuring of the output of the nanomachine. The buffer solutions used for all the experiments are already reported in[23] and contained physiological concentrations of ATP (2 mM) unless differently specified. 0.5% methyl-cellulose was added to the running buffer in order to inhibit the lateral diffusion of F-actin[37] and minimise the probability of loss of acto–myosin interaction. The concentration of HMM from soleus and psoas muscle used for the experiments was defined by the concentration at which the number of rupture events in rigor attained a saturating value.

The mechanical apparatus, as already reported in[23], can be operated either in position clamp (Fig. 2, red branch), when the feedback signal is the position of the nanopositioner carrying the motor array ($x$), or in force clamp (green branch), when the feedback signal is the force ($F$), calculated as the product of the stiffness of the trap ($e$) times the change in position of the bead in the laser trap ($x_{bead}$). Recording of the nanomachine performance in true isometric condition, however, cannot be achieved in position clamp, due to the large trap compliance ( ~ 4 nm pN$^{-1}$), which implies both several tens of nanometres movement to develop the maximum steady force and blunting of the force of individual attachment–detachment events (Supplementary Fig. 7 in[23]). To eliminate the trap compliance the system has been implemented with a length clamp (blue branch in Fig. 2), which uses as a feedback signal the change in distance ($L$) between the position of the actin attached bead in the laser trap ($x_{bead}$) and that of the nanopositioner ($x$), so that the movement of the bead with the force change is counteracted by the movement of the nanopositioner. In this way the effective trap compliance is reduced to 0.2 nm pN$^{-1}$.

In length clamp the frequency response of the system is reduced by the propagation time of the mechanical signal through the loop from the force transducer to the nanopositioner, which also includes the array of actin attached myosin motors. The power density spectrum (PDS) of the system, measured with sinusoidal oscillations at different frequencies, changes depending on the selected feedback mode: in position clamp the PDS shows an upper − 3 dB frequency (or corner frequency $f_c$) of 59 Hz (Fig. 6, red); in

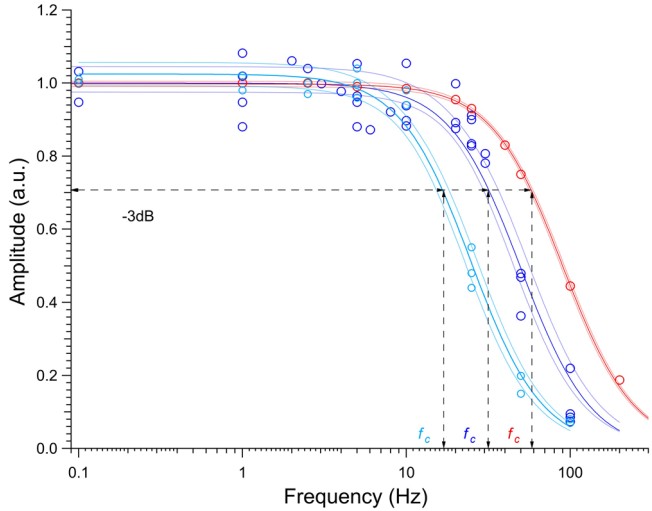

**Fig. 6 | Power density spectrum of the system.** Superimposed power density spectrum either in position clamp (red circles interpolated by the red Lorentzian curve), or in length clamp with the array of actin attached motors in rigor from both fast muscle (violet circles and curve) and slow muscle (cyan circles and curve). The upper $-3$ dB frequency $f_c$ is: $59 \pm 3$ Hz (red), $31 \pm 6$ Hz (violet) and $17 \pm 3$ Hz (cyan). The coloured area delimited by thinner lines indicates the confidence limits.

length clamp, when the feedback loop is closed with the array of actin–attached myosin motors in rigor, $f_c$ decreases to 32 Hz with HMM from fast muscle (violet) and to 17 Hz with HMM from slow muscle (cyan). The mass of the system ($m$) is the same with either isoform array thus the different corner frequency between the two nanomachines should almost in part depend on the different stiffness of the two arrays in rigor.

The architecture of the machine (with the length of the motor array much shorter than the length of the overlapping actin filament) implies that, for a given HMM concentration, the measured number of rupture events does not significantly change from experiment to experiment, therefore there is no need to normalise the mechanical response obtained in different experiments at physiological [ATP] by the actin–filament length. All the experiments were conducted at room temperature (24 °C).

## Statistics and reproducibility
Data are expressed as mean ± standard deviation unless otherwise stated. The number of replicates is defined in the text and in the figure legends.

## Stochastic model: on the governing master equation
The master equation can be cast in the general form:

$$\frac{\partial P(\boldsymbol{n}, t)}{\partial t} = \sum_{\boldsymbol{n}' \neq \boldsymbol{n}} [T(\boldsymbol{n}|\boldsymbol{n}')P(\boldsymbol{n}', t) - T(\boldsymbol{n}'|\boldsymbol{n})P(\boldsymbol{n}, t)] \quad (3)$$

where $T(\boldsymbol{n}'|\boldsymbol{n})$ represent the transition rates from the state $\boldsymbol{n}$ to a new state $\boldsymbol{n}'$, compatible with the former. In the following, to identify the arrival/departure state $\boldsymbol{n}'$ we solely highlight the individual component of the vector $\boldsymbol{n}$ that changes due to the considered reaction. The explicit expression for the transition rates that originates from the chemical equations (1) is given in the annexed Supplementary Note 1.

In Supplementary Note 1 are also presented the details of the numerical simulation of a single stochastic orbit of the considered dynamics, obtained via the celebrated Gillespie algorithm[38,39]. In Supplementary Fig. 3 it is shown the time evolution of the (discrete) concentration of molecular motors in each configuration, while in Supplementary Fig. 4 the results of the stochastic simulations for the probability distributions of the fractions of motors in the force-generating configurations are compared with the theoretical prediction.

## The deterministic limit
From the master equation one can readily derive the mean field equations that govern the deterministic dynamics for the continuous concentrations of the molecular motors in configurations $A_1$ and $A_2$:

$$\begin{cases} \frac{dy}{dt} = k_1 - (k_1 + k_{-1} + k_2)\, y - (k_1 - k_{-2})\, z \\ \frac{dz}{dt} = k_2\, y - (k_{-2} + k_3)\, z \end{cases} \quad (4)$$

where $y$ and $z$ identify the averaged fraction of the molecular motors in states $A_1$ and $A_2$, respectively. Equations (4) can be studied at equilibrium, yielding the fixed point solutions:

$$\begin{cases} y^* = \left(\frac{k_1}{k_1 + G}\right) \frac{k_{-2} + k_3}{k_2 + k_{-2} + k_3} \\ z^* = \left(\frac{k_1}{k_1 + G}\right) \frac{k_2}{k_2 + k_{-2} + k_3} \end{cases} \quad (5)$$

where:

$$G = \frac{k_{-1}(k_{-2} + k_3) + k_2 k_3}{k_2 + k_{-2} + k_3} \quad (6)$$

We define the duty ratio $r$ as the fraction of attached motors (or the fraction of the ATPase cycle time a motor spends attached). It can be computed as:

$$r = y^* + z^* = \frac{k_1}{k_1 + G}. \quad (7)$$

The temporal evolution of the mean field concentrations of motors in different configurations, for a suitable choice of the kinetic parameters is shown in Supplementary Fig. 2, in Supplementary Note 1. A straightforward calculation can be performed to show that $z^* \simeq r = \frac{k_1}{k_1 + G}$ and $y^* \ll 1$, when $k_{-2}/k_2, k_3/k_2 \ll 1$. In practical terms, under this operating assumption, which for the mammalian muscle myosin under consideration is approached at temperature $T \simeq 24$ °C, motors are solely found in state $A_2$. As discussed earlier, this is the relevant setting for the specific case study at hand.

Equations (4) can be drastically simplified by performing a self-consistent elimination of the variable $y$. To this end we set $dy/dt = 0$ in the first of equations (4) to eventually express $y$ as a function of $z$. This procedure is customarily invoked to carry out the so-called adiabatic elimination, which proves correct when there is a clear separation of time scales between co-evolving variables. Although this is not a priori the case for the system at hand, we will postulate the validity of the aforementioned condition and operate with the ensuing approximation that, as we shall prove, will materialise in an accurate interpretative framework. Further details can be found in Supplementary Note 1 (see Supplementary Fig. 1). Plugging the expression for $y$ as a function of $z$ into the second of equations (4) and solving the ensuing differential equation readily yields:

$$\frac{dz}{dt} = \frac{k_1 k_2}{k_1 + k_{-1} + k_2} - z\left[k_{-2} + k_3 + \frac{k_2(k_1 - k_{-2})}{k_1 + k_{-1} + k_2}\right] \equiv b - az \quad (8)$$

which immediately yields solution (2), as reported in the Results. From equation (2) we can write $z^* = b/a$ and this latter condition matches the homologous estimate derived from the original two-dimensional model and reported in equations (5).

## Exact solution of the stochastic problem
At first, we remark that the probability distribution $P(\boldsymbol{n}; t) \equiv P(n_1, n_2; t)$ can be written as a vector $\boldsymbol{P}(t)$ of dimension $(N + 1) \times (N + 1)$. This latter returns the probability at time $t$, of finding the system in the state characterised by $n_1$ motors in configuration $A_1$ and $n_2$ motors in configuration $A_2$

$A_2$. Here, $n_1$ and $n_2$ can in principle assume every integer values in the range $[0, N]$, i.e. a total of $N + 1$ values each. Observe however that the populations of motors in the configurations $A_1$ and $A_2$, must satisfy the obvious constraint that reflects the conservation law, i.e. $n_1 + n_2 \leq N$. A simple way to express the condition above is to consider that for each possible value of $n_1$, $n_2$ can assume values in the range $[0, N - n_1]$. This readily implies that the total number of possible states for the system is identically equal to $M = (N + 1)(N + 2)/2$. The number of allowed states are hence smaller than what anticipated above. Indeed the non trivial entries of $\boldsymbol{P}(t)$ are $M = (N + 1)(N + 2)/2$. We will consequently focus on the non trivial elements of vector $\boldsymbol{P}(t)$ which we shall denote as $P_m(t)$ with $m = 1, …, M$. For the relevant case of $N = 16$ molecular motors, instead of $(N + 1) \times (N + 1) = 289$ configurations we only have $M = 153$ possible states that can be eventually visited by the system, and that we explicitly list in Supplementary Note 1. As we shall also discuss in the same Supplementary Note 1, the stationary probability distribution $\boldsymbol{P}^{st}$ defines the kernel of a $M \times M$ matrix $\mathbb{Q}$ and can be hence computed as the eigenvector of $\mathbb{Q}$ relative to the null eigenvalue. The entries of the matrix $\mathbb{Q}$ can be computed from the transition rates of the underlying master equation, as highlighted in Supplementary Note 1.

The marginal probability $\rho_k$ to find $k \leq N$ motors in $A_2$ can be extracted from the stationary probability distribution $\boldsymbol{P}^{st}$, the stationary solution of the master equation. This is done by summing the elements of $\boldsymbol{P}^{st}$ that refer to the selected $k$, and that account for all possible partitioning of the remaining $N - k$ motors among configurations $D$ and $A_1$. The knowledge of the stationary probabilities ($\rho_0, \rho_1, \rho_2, …, \rho_N$) opens up the perspective to calculate the stationary state distribution of the applied force $F$, an important asset when aiming at a refined fitting scheme which meticulously accounts for the role played by fluctuations.

To work along these lines we shall assume that the contribution to the force (including fluctuations) of the motors in the state $A_1$ is always negligible. This assumption is motivated by the fact that, for the experimental setting here explored, only a tiny fraction of motors is found to populate state $A_1$, at any time $t$. In Supplementary Note 1, we will however relax this working assumption so as to provide a rigorous theoretical framework that extends to account for the relevant setting where the population of $A_1$ motors is instead significant in size.

Let us focus on $k \leq N$ distinct motors in state $A_2$. As postulated earlier, each motor can exert a constant random force $f$, uniformly spanning the assigned interval $\mathcal{I}_2$. For each choice of $k$, we can compute the distribution of the forces $\Pi_k(f)$ applied by the selected $k$ motors, by combining independent and identically uniformly distributed random variables drawn for the interval of pertinence $\mathcal{I}_2$ (see Supplementary Note 1 for further technical information, Supplementary Figs. 6 and 7). As stated in the Results, functions $\Pi_k(f)$ need to be combined together with proper weighting factors that reflect the stationary probability $\rho_k$ of having exactly $k$ motors in the force-generating state $A_2$, namely $P(F) = \sum_{k=0}^{N} \rho_k \Pi_k(f)$.

The relevant steps of the fitting strategy are listed below. We begin by focusing on the average force profile, hence disregard the impact of finite size fluctuations. As mentioned above, the time evolution of the recorded force can be approximated by an effective, two-parameters model (see Supplementary Fig. 5 in Supplementary Note 1). The latter parameters – respectively denoted $F_0$ and $a$ – can be estimated via a direct fit. Having accessed to preliminary estimated values for the average force at the stationary plateau $F_0$ and for the rate of isometric force development $a$, one can then set forth to characterise the other kinetic parameters by analysing the distribution of the fluctuations of the force around the asymptotic plateau. To this end we note that $f_0$, one of the unknown of the model, can be written as:

$$f_0 = \frac{20}{11} \frac{F_0}{N} \frac{a}{b} \tag{9}$$

where $a$ is constrained to the value determined above while $b = k_1 k_2 / (k_1 + k_{-1} + k_2)$ as defined by equation (8).

Armed with the above knowledge, we can proceed further by comparing the probability density function of the force fluctuations as obtained analytically to the homologous histogram computed from the stochastic simulations. The former is adjusted to the latter by modulating the free parameters $k_1$, $k_{-1}$, $k_2$, $k_{-2}$ and $k_3$, for a fixed choice of $N$. As discussed in Supplementary Note 1, testing the method against synthetic data generated in silico enables us to conclude that parameters $f_0$ and $r = k_1/(k_1 + G)$ can be correctly estimated, following the above fitting scheme (see Supplementary Table 2). Also the estimated $b$ and $a$ (recomputed from the best fitted values for the kinetic constants) are pretty close to their nominal values as imposed in the simulations. Remarkably $\phi$, the rate of motors completing the interacting cycle with the actin, is also correctly recovered. A graphic comparison between estimated and exact parameters (i.e. those employed in the inspected simulations) is also shown in Supplementary Fig. 8 of Supplementary Note 1. Notice however that multiple combinations of the parameters $k_1$, $k_{-1}$, $k_2$, $k_{-2}$ and $k_3$ exists that yields the same fitted profile (with almost identical estimates for the relevant quantities $f_0$, $r$, $a$ and $b$).

While the kinetics of the scrutinised model cannot be solved unequivocally, we are in a position to accurately determine crucial parameters – as e.g. the maximum force exerted by a single motor and the associated duty ratio of the ensemble – which proved elusive under the deterministic viewpoint, as can be appreciated in Supplementary Fig. 9 of Supplementary Note 1.

The above analysis refers to a fixed value of $N$, the size of the system that we assumed (from the experiment results shown in Fig. 3)$N = 16$. In principle the correct value of $N$ is not a priori known. To overcome this intrinsic limitation, one could repeat the analysis by varying $N$ and recording the parameter estimated as follows the fitting scheme. Here, we will consider the simplified setting where $a$ and $b$ are frozen to the values determined for $N = 16$ (so that $z^*$ stays unchanged). This is implemented by removing two parameters from the pool of quantities to be fitted. Specifically $k_{-1}$ and $k_3$ are constrained to match two constitutive relations, that involve $k_1$, $k_2$ and $k_{-2}$, in addition to $a$ and $b$. The parameters to be fitted are hence $k_1$, $k_2$ and $k_{-2}$, while $k_{-1}$, $k_3$ and $f_0$ can be self-consistently determined from the their best fit values. Notice that $f_0$ is expected to change as a function of $N$ as specified by relation (9). The result of the analysis are reported in Supplementary Fig. 10 in Supplementary Note 1: the fitting procedure converges (with the requested limit of precision) only over a finite range of values of $N$, centred around the value adopted when performing the simulations. This observation implies in turn that we are in a position to obtain a reasonable estimate for the interval of pertinence of $N$, as follows the procedure outlined above.

The introduced theoretical framework and the ensuing fitting strategy, thoroughly validated against synthetic data, can be hence applied to the analysis of the experimental data so to yield a self-consistent estimate of the underlying mechanokinetic parameters. For a detailed validation of the fitting scheme against synthetically generated data refer to Supplementary Note 1.

## Data collection and analysis

A custom built program written in LabVIEW (National Instruments) was used for signal recording. Data analysis was carried out using LabVIEW (National Instruments) and MATLAB (MathWorks) dedicated scripts, and Origin 2018 (OriginLab Corp., Northampton, MA, USA) and Igor Pro 8 (WaveMetrics, Portlan, OR, USA) software.

## Reporting summary

Further information on research design is available in the Nature Portfolio Reporting Summary linked to this article.

## Data availability

The source data for all figures and tables are provided as Supplementary Data 1. All remaining data will be available from the corresponding authors upon reasonable request.

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

## Acknowledgements

We thank Gabriella Piazzesi for critical evaluation of this work. We thank the staff of the mechanical workshop of the Department of Physics and Astronomy (University of Florence) for mechanical engineering support. This work was supported by Fondazione Cassa di Risparmio di Firenze (2020.1582), Italy, the Italian Ministry of Education, Universities and Research (PRIN2020, CUP: B55F22000540001), the European Joint Program on Rare Diseases 2019 (PredACTINg, EJPRD19–033), and the Next Generation EU programme [DM 1557 11.10.2022] in the context of the National Recovery and Resilience Plan, Investment PE8–Project Age–It: "Ageing Well in an Ageing Society", Italy.

## Author contributions

P.B., V.L. and D.F. designed the research. I.P., P.B., M.L., M.C., M.M., and I.M. performed the experiments and analysed the data. V.B. and D.F. developed the stochastic model. V.L., D.F., V.B., P.B. and M.R. wrote the paper or revised it critically for important intellectual content. All authors participated in discussions on this work and approved the final version of the manuscript.

## Competing interests

The authors declare no competing interests.
