## [Peer Review File · Communications Biology]

Reviewers' comments:

Reviewer #1 (Remarks to the Author):

This paper presents force measurements of eight molecules of fast or slow myosins interacting with actin using their originally developed feedback-based system, and developed an analytical model of acto-myosin interaction that reproduces the experimental data. To gain further insight into the molecular mechanism by which the isometric tension cost in fast muscle is 3-5 times higher than in slow muscle, the key molecular parameters, such as the force of a motor, duty ratio, and rate of transition during the attachment-detachment cycle estimated by this model are discussed. The experimental method is innovative, incorporating feedback control, and thus, the authors have succeeded in suppressing the adverse effects on myosin-actin interactions caused by the trap compliance. It is also worth noting that the parameters estimated by the analytical model based on the minimum mechanochemical state for the steady-state contraction, are almost equivalent to those estimated by the Monte Carlo-based numerical simulation model. Thus, this paper is of interest to others in this community and deserves to be published in this journal once the following justifications are made. Particularly, I am not fully satisfied with the following aspects: a) the arguments about the isometric tension cost of fast-twitch muscle being about 3-5 times higher than that of slow-twitch muscle, and b) clear descriptions for estimating the molecular parameters using the analytical model.

1) In the introduction, the isometric tension of fast and slow muscles is almost the same or 1.5 times higher in fast muscle, but the ATP hydrolysis rate per motor is 5 times higher in fast muscle, suggesting the need to consider the molecular mechanism by which the isometric tension cost is 5 times higher in fast muscle. The results of the present experiment and the estimated results, however, are not sufficient to elucidate the molecular mechanism of higher isometric tension cost in fast-twitch muscle. The potential explanations for these discrepancies are discussed in the first paragraph of Page 12, but a little more should be said about the validity of the results of this study. For example, it has been reported that the force of slow myosin is three times smaller than that of fast myosin, albeit estimated from muscle fibers (Percario et al. 2017 J Physiol), and these results are consistent with those of this study. These discrepancies and consistencies should be discussed further, as well as how the different molecular properties of fast and slow myosins contribute to their distinctive function in the slow and fast muscles.

2) With regard to the descriptions and the validation of the analytical model, it took me an enormous amount of time just to get a rough understanding of it. If this model is to be promoted in this field, it needs to be rewritten to make it easier for the reader to understand. In particular, to understand the procedure for estimating the three molecular parameters, f_0 , r , and ϕ , I had to read the main text, the methods section, and the Supplementary Information in detail and spend an enormous amount of time piecing this information together. Below is the procedure as I understand it, although I am still not certain if this is correct. Such clear descriptions of the estimation procedure should be described in the Methods section, but the current explanation would be incomprehensible to most readers. Significant modifications are needed.

The following is my best understanding of the steps in the parameter estimates:

Step1) In the master equation as shown in equation (3), the stationary probability distribution P^{st} is obtained for the steady-state situation.

Step2) Find the stationary probability p_k by summing the stationary probability distribution P^{st} of the k motors.

Step3) Find the probability distribution of force fluctuation Π by Suppl Info equation (7).

Step4) The distribution of force fluctuations (Suppl Fig. 7) is calculated by the sum of the products of p_k and Π , and the rate constants $k_1 \sim k_3$ are estimated to fit the experimentally obtained force distribution with the estimated distributions (Suppl Fig. 11).

Step5) Parameters, a and F_0 , are obtained from experimental fits, b is calculated as a function of rate constants, $k_1 \sim k_3$, and finally key parameters are estimated as follows, $r=b/a$, $f_0=20/11*F_0/N/r$, $\phi=k_1(a-b)/(k_1+G)$.

Specific and minor comments:

1) Abstract:

In the sentence "... the slow muscle (responsible for the posture) ...", it has been reported that slow muscles work as a voluntary muscle more frequently than fast muscles for slow locomotion such as walking. Therefore, it should be modified to be "primarily responsible for the posture".

2) Page2, Line8 from bottom

Same as the previous comment (1), please modify as follows: "Slow muscles, which are involved primarily in maintenance of posture ..."

3) Page5, 2nd paragraph

This paragraph describes a series of experiments in which force is measured by two types of feedback modes, position clamp and length clamp. It will be easier for reader to understand why the force measurements were made with position clamp first, followed by length clamp before describing the details of the experiment as shown in Fig.4 on page6.

I also have a fundamental question: Why is the increase in the noise superimposed on the steady force expected in length clamp?

4) Supplementary Information Page13, L7

Typo: "a s well as the value ..." should be "as well as the value ..."

5) Supplementary Information Table2 and Fig.8

The parameter values from the numerical simulation model are referred to as "True parameters," but since these are also "estimated" parameters, I am not convinced by the term "True".

Reviewer #2 (Remarks to the Author):

In this study, authors used a unidimensional synthetic nanomachine powered by pure myosin isoforms from rabbit skeletal muscles to extract mechanokinetic parameters that underlie the force generation in both slow and fast muscles. By fitting experimental data to a theoretical model, they estimated various mechanokinetic properties of the motor ensemble, such as the motor force, the fraction of actin-attached motors, and the rate of transition through the attachment-detachment cycle.

This study holds significant interest and importance as it paves the way for future research on the emergent mechanokinetic properties exhibited by ensembles of myosin molecules. This work can be accepted after following concerns are appropriately addressed.

Main concerns:

1. How does the structure of unidimensional synthetic nanomachines primarily differ from that of a sarcomere? For instance, a sarcomere in skeletal muscles consists of one actin filament symmetrically surrounded by three thick filaments, facilitating straight movement of the thin filament. Is the experimental setup, including the motor ensemble and actin filament, truly unidimensional?
2. What about the time resolution, space resolution, and force resolution in the experiments?
3. Is the kinetic scheme given in Eq. (1) appropriate? For some motors that are not properly oriented, they may not be able to attain the high-force generating state. Indeed, this may explain the lower ϕ obtained in experiments.
4. The modeling aspect is crucial in extracting physical parameter values in this study. The emergent behavior of an ensemble of myosins is not solely represented by their average behavior. Some coordination among motors, for example, through force, should be considered in the theoretical part, which we are unable to find. The rates for different states in the model seem to have been set as constants in this work.

Minor concerns:

5. Page 1. "In isometric contraction,the lever arm". There exists a grammar error in this sentence.
6. There are quite a few grammar errors in the paper. Please carefully correct them.
7. It will be good if the description of the theoretical part gets clearer.

Reviewer #3 (Remarks to the Author):

The authors present a combined experimental-theoretical study of force generation by fast (from psoas muscle) versus slow (from soleus muscle) muscle myosin II isoforms. The myosin II molecules are absorbed onto a micropipette and attached to an actin filament held in an optical trap, which allows to measure force (setup introduced earlier in Ref. 23). A calibration experiment yields that $N=16$ motor heads are building up this "nanomachine". Each of these motors is now assumed to be in one of three states (dissociated, weakly attached or post powerstroke). The corresponding master equation is attacked in many different ways, including moment (or mean field) equations, an adiabatic approximation and stochastic simulations with the Gillespie algorithm. Fitting to the experimental results leads to the main results of this study, namely the parameters reported in Table 1. The fast isoform generates a larger force f_0 , has a smaller duty ratio r and a larger dissociation rate ϕ . Because this work presents results for both fast and slow isoforms, it is an advance over Ref. 23, which was only for the fast version. The results are also in good agreement with work from the Yanagida's group (Ref. 22, BPJ 1996).

This work is interesting, novel and solid, but highly specialized. It is not clear if it of large interest for the general reader of this journal; it might be more suited for a journal on muscle physiology. To make it more accessible, it should be rewritten such that one can follow it more easily. Technical details should go into the supplement and the text should focus on the essential message. For example, it does not make sense to use equations in the main text that appear only later in the methods section. Either these equations are in the main text, too, or the main text can do without them. Similar with the "nanomachine": it is introduced rather late and without much emphasis, as if the reader had to know it. When it is finally explained in Fig. 2, then in a very technical manner, with a focus on the control structure. Given its importance (and also the title of the manuscript), it should be introduced rather at the very beginning and with clear explanations regarding its nature and significance.

Apart from these concerns regarding general interest and accessibility, I also have a few more scientific concerns. The name "nanomachine" suggests a well-defined setup, but if I understand correctly, the number of motors is determined by an adsorption process at a certain concentration. Why can the authors assume that always around 16 heads are active? What makes their setup so deterministic? Why should these 16 heads be all equivalent? I can imagine that interactions with the actin would be very variable along the filament. I could imagine other setups, e.g. with DNA-origami, which are much more controlled (compare e.g. Derr, Nathan D., et al. "Tug-of-war in motor protein ensembles revealed with a programmable DNA origami scaffold." *Science* 338.6107 (2012): 662-665). For me this setup appears close to the traditional three-bead setup from Ref. 21. Also, there are many other works of this kind with motor ensembles not cited here, e.g. Debold, Edward P., et al. "Direct observation of phosphate inhibiting the force-generating capacity of a miniensemble of myosin molecules." *Biophysical journal* 105.10 (2013): 2374-2384.

A second major issue is that the rates of the model seem not to depend explicitly on ATP-concentration and force. Should k_3 not be linear in ATP-concentration? And k_1 and k_2 mechanosensitive? Why do the authors not make use of this opportunity to fit to experimental data? I would have expected systematic variation of ATP-concentration. Can this be added? Very importantly, I am missing load sharing, which adds a strongly non-linear aspect to the system and can lead to rupture cascades, as actually observed here. As an example for a very similar model that implemented these features, I mention the parallel cluster model by Erdmann, Thorsten, and Ulrich S. Schwarz. "Stochastic force generation by small ensembles of myosin II motors." *Physical review letters* 108.18 (2012): 188101. Why can the authors here neglect that force has to be shared between the different motor heads?

Reply to Reviewers

We have carefully revised the text (edited text in red) according to the criticisms and suggestions of the reviewers, as detailed point by point in this reply.

Reply to Reviewer #1

This paper presents force measurements of eight molecules of fast or slow myosins interacting with actin using their originally developed feedback-based system, and developed an analytical model of acto-myosin interaction that reproduces the experimental data. To gain further insight into the molecular mechanism by which the isometric tension cost in fast muscle is 3-5 times higher than in slow muscle, the key molecular parameters, such as the force of a motor, duty ratio, and rate of transition during the attachment-detachment cycle estimated by this model are discussed. The experimental method is innovative, incorporating feedback control, and thus, the authors have succeeded in suppressing the adverse effects on myosin-actin interactions caused by the trap compliance. It is also worth noting that the parameters estimated by the analytical model based on the minimum mechanochemical state for the steady-state contraction, are almost equivalent to those estimated by the Monte Carlo-based numerical simulation model. Thus, this paper is of interest to others in this community and deserves to be published in this journal once the following justifications are made. Particularly, I am not fully satisfied with the following aspects: a) the arguments about the isometric tension cost of fast-twitch muscle being about 3-5 times higher than that of slow-twitch muscle, and b) clear descriptions for estimating the molecular parameters using the analytical model.

1) In the introduction, the isometric tension of fast and slow muscles is almost the same or 1.5 times higher in fast muscle, but the ATP hydrolysis rate per motor is 5 times higher in fast muscle, suggesting the need to consider the molecular mechanism by which the isometric tension cost is 5 times higher in fast muscle. The results of the present experiment and the estimated results, however, are not sufficient to elucidate the molecular mechanism of higher isometric tension cost in fast-twitch muscle. The potential explanations for these discrepancies are discussed in the first paragraph of Page 12, but a little more should be said about the validity of the results of this study. For example, it has been reported that the force of slow myosin is three times smaller than that of fast myosin, albeit estimated from muscle fibers (Percario et al. 2017 J Physiol), and these results are consistent with those of this study. These discrepancies and consistencies should be discussed further, as well as how the different molecular properties of fast and slow myosins contribute to their distinctive function in the slow and fast muscles.

The suggestion of the reviewer has been taken into account in Discussion, revising the question in terms of the mechanochemical cycle shown in Fig. 1 and taking into account the relevant literature on both the mechanical and kinetic parameters implied.

2) With regard to the descriptions and the validation of the analytical model, it took me an enormous amount of time just to get a rough understanding of it. If this model is to be promoted in this field, it needs to be rewritten to make it easier for the reader to understand. In particular, to understand the procedure for estimating the three molecular parameters, f_0 , r , and φ , I had to read the main text, the methods section, and the Supplementary Information in detail and spend an enormous amount of time piecing this information together. Below is the procedure as I understand it, although I am still not certain if this is correct. Such clear descriptions of the estimation procedure should be described in the Methods section, but the current explanation would be incomprehensible to most readers. Significant modifications are needed. The following is my best understanding of the steps in the parameter estimates:

Step1) In the master equation as shown in equation (3), the stationary probability distribution P^{st} is obtained for the steady-state situation.

Step2) Find the stationary probability ρ_k by summing the stationary probability distribution P^{st} of the k motors.

Step3) Find the probability distribution of force fluctuation Π by Suppl Info equation (7).

Step4) The distribution of force fluctuations (Suppl Fig. 7) is calculated by the sum of the products of ρ_k and Π , and the rate constants $k_1 \sim k_3$ are estimated to fit the experimentally obtained force distribution with the estimated distributions (Suppl Fig. 11).

*Step5) Parameters, a and F_0 , are obtained from experimental fits, b is calculated as a function of rate constants, $k_1 \sim k_3$, and finally key parameters are estimated as follows, $r=b/a$, $f_0=20/11 * F_0/N/r$, $\varphi=k_1(a-b)/(k_1+G)$.*

The description of the theoretical part has been modified so as to reflect the referee's remarks. First of all we have removed technical references to the adiabatic elimination procedure, and relegated the discussion to the Method section. We also avoid referring in the main text to equations that appear later on (as also requested by Referee 3). More importantly, and following the synthetic (and correct!) outline provided by the referee, we have inserted in the revised version of the paper a punctuated list of the successive steps involved in the theoretical analysis to help the reader grasping the essence of the employed fitting strategy. We hope that with this revised version of the text we have successfully addressed the referee's concern.

Specific and minor comments:

1) Abstract: In the sentence "... the slow muscle (responsible for the posture) ...", it has been reported that slow muscles work as a voluntary muscle more frequently than fast muscles for slow locomotion such as walking. Therefore, it should be modified to be "primarily responsible for the posture".

Done.

2) Page2, Line8 from bottom Same as the previous comment (1), please modify as follows: "Slow muscles, which are involved primarily in maintenance of posture ..."

Done.

3) Page5, 2nd paragraph This paragraph describes a series of experiments in which force is measured by two types of feedback modes, position clamp and length clamp. It will be easier for reader to understand why the force measurements were made with position clamp first, followed by length clamp before describing the details of the experiment as shown in Fig.4 on page6. I also have a fundamental question: Why is the increase in the noise superimposed on the steady force expected in length clamp?

The reviewer is right in noting the insufficient justification of the measurements in position and length clamp and their sequence. Following the reviewer question, we anticipated at page 3 a synthetic explanation of the length clamp and the rationale for the need of the length clamp in our investigation. The reason for starting the experiment in position clamp (PC) and switching to length clamp (LC) only after the establishment of a steady interaction is purely technical and is now made explicit while describing the experiment at page 5. The increase in force noise with the switch PC-LC is the consequence of the change in the effective compliance in series with the motor system. In PC any new force generating interaction is dissipated in filament sliding against the large trap compliance (4 nm/pN); in LC the equivalent series compliance is reduced to 0.2 nm/pN, so that the addition-subtraction of force by any attachment-detachment is preserved. This point is now made clear in the text at page 5. We thank the reviewer for promoting all these clarifications.

4) Supplementary Information Page13, L7 Typo: "a s well as the value ..." should be "as well as the value ..."

Done.

5) Supplementary Information Table2 and Fig.8 The parameter values from the numerical simulation model are referred to as "True parameters," but since these are also "estimated" parameters, I am not convinced by the term "True".

With “True parameters” we refer to the parameters that are imposed in the simulation (and thus known with infinite precision) and that we wish at recovering (or estimating) via the proposed fitting strategy. A sentence has been added in the Supplementary Information to clarify this point.

Reply to Reviewer #2

In this study, authors used a unidimensional synthetic nanomachine powered by pure myosin isoforms from rabbit skeletal muscles to extract mechanokinetic parameters that underlie the force generation in both slow and fast muscles. By fitting experimental data to a theoretical model, they estimated various mechanokinetic properties of the motor ensemble, such as the motor force, the fraction of actin-attached motors, and the rate of transition through the attachment-detachment cycle. This study holds significant interest and importance as it paves the way for future research on the emergent mechanokinetic properties exhibited by ensembles of myosin molecules. This work can be accepted after following concerns are appropriately addressed.

Main concerns: 1.

How does the structure of unidimensional synthetic nanomachines primarily differ from that of a sarcomere? For instance, a sarcomere in skeletal muscles consists of one actin filament symmetrically surrounded by three thick filaments, facilitating straight movement of the thin filament. Is the experimental setup, including the motor ensemble and actin filament, truly unidimensional?

In a previous paper (Pertici et al., *Nat Commun.* 2018 (ref. 23)) we described all the technical features of the nanomachine powered by fast myosin from skeletal muscle and discussed in detail the differences with the performance of the native half-sarcomere of the fast muscle. The comparative analysis was possible exploiting a model simulation based on a simplified mechano-kinetic scheme with parameters that were first selected to fit the known mechanical and energetic performance of the muscle at the level of the half-thick filament (300 motors able to interact with the nearby actin filaments). Then the same mechano-kinetic scheme was scaled down to the level of the unidimensional nanomachine, taking into considerations the two intrinsic methodological limits of the nanomachine: (i) the random orientation of the myosin motors, which on average depresses the force of the interaction (Pertici et al., *Nat Commun.* 2018 (ref. 23); Ishijima et al., *Biophys J* 1996 (ref. 22)) and decreases low load step size (Pertici et al., *Int J Mol Sci* 2020 (ref. 24); Tanaka et al., *Biophys J* 1998, 75, 1886-1894); (ii) the two orders of magnitude larger in series compliance (due to the trap compliance in position clamp) with respect to the *in situ* experiments (ref. 23). Under these conditions it was found that the relevant mechanical parameters of the nanomachine (the isometric force, the force-velocity relation and thus the power) were fitted by the kinetic scheme selected to fit the *in situ* performance, when the number of motors available for the interaction with the actin filament was scaled down to 16, in astonishing agreement with the number of motors estimated from the rupture experiment in rigor. This demonstrated the reliability and the limits of the nanomachine as a reduced model of the half-sarcomere.

2. What about the time resolution, space resolution, and force resolution in the experiments?

The dynamic ranges of force and movement of the DLOT-nanopositioner system are given in all previous publications on the nanomachine and reported also here in the first paragraph of the section of Methods “Mechanical experiments”. The frequency response of the system varies according to the mode of operation (position clamp, length clamp) as detailed in the section “Mechanical experiments” of Methods and in the related Fig. 6.

3. Is the kinetic scheme given in Eq. (1) appropriate? For some motors that are not properly oriented, they may not be able to attain the high-force generating state.

The reviewer observation is quite important. In fact, in our previous published work on the nanomachine (refs. 23-24) we carefully analysed the problem. As anticipated in the reply to main concern 1, the random orientation of motors is one of the two intrinsic limits of the nanomachine (the other being the two orders of magnitude larger series compliance) that must be taken into account for a meaningful comparison with *in situ* performance of the fast myosin array. The consequences of these two limits for the performance of the nanomachine in comparison with *in situ* performance were analysed in detail by model simulation (refs. 23-24). As far as the effect on the isometric force and the related energetics the results are summarized in Table 1 of the first paper, which is reported here for clarity:

Table 1 Simulated mechanical and energetic parameters of the half-sarcomere and their modulation by the conditions imposed by the nanomachine.

	F_0 (pN)	a/F_0	P_{max} (aW)	φ_0 (s ⁻¹ per head)	$\varphi_{P_{max}}$ (s ⁻¹ per head)	$r_{s,0}$ (s ⁻¹ per head)	$r_{s,P_{max}}$ (s ⁻¹ per head)
Compliance 0.01 nm·pN ⁻¹	433 ± 5	0.365	462	11.65	35.50	0.60	31.24
Compliance 3.7 nm·pN ⁻¹	552 ± 1	0.234	437	14.50	33.82	15.60	23.87
Compliance 3.7 nm·pN ⁻¹ + random	312 ± 7	0.188	237	14.48	33.70	13.52	24.63

F_0 : force per half-thick filament; a/F_0 : relative value of the parameter a of Hill's hyperbolic equation²¹; P_{max} : maximum power; φ : flux through step 1 of the cycle in Fig. 4a, corresponding to the ATP hydrolysis rate either in isometric condition (φ_0) or at P_{max} ($\varphi_{P_{max}}$); r_s : slipping rate within the same ATPase cycle (step "slip" in Fig. 4a) in isometric condition ($r_{s,0}$) and at P_{max} ($r_{s,P_{max}}$). Upper row: *in vivo* series compliance; middle row: trap compliance; lower row: trap compliance with random orientation of motors.

The two orders of magnitude larger compliance of the nanomachine (second row) with respect to the native in series compliance (first row) causes that each addition-subtraction of force induces substantial sliding undermining the condition of independent force generators peculiar to the motors in the half-sarcomere array. Consequently, the strain-dependent kinetics of the attached heads also in the isometric condition in position clamp is influenced by the push-pull experienced by all the other attached heads when actin slides away-toward the bead for the addition-subtraction of the force contribution by one head (see ref. 23, Supplementary Fig. 7). The isometric ATPase rate (φ_0) increases accordingly (compare the values in the first and second row). Notably the large compliance effect on φ_0 remains dominant also when the average force per motor is reduced by the assumption of random orientation (compare second and third row).

Indeed, this may explain the lower φ obtained in experiments.

Please note that the simulated isometric φ (φ_0) is obtained in the paper under revision from fits applied to force fluctuations in length clamp conditions and thus with the motors acting effectively in parallel as independent force generators. Thus, the stochastic model used in this paper does not need to include the series compliance. Instead, the model **does include** the assumption of the reduction of the average motor force due to random orientation of motors.

4. The modeling aspect is crucial in extracting physical parameter values in this study. The emergent behavior of an ensemble of myosins is not solely represented by their average behavior. Some coordination among motors, for example, through force, should be considered in the theoretical part, which we are unable to find. The rates for different states in the model seem to have been set as constants in this work.

As detailed in the reply above, with the length clamp that reduces the series compliance to a value comparable to that *in situ*, we were able to approach the condition of motors in parallel acting as independent force generators.

Minor concerns:

5. Page 1. "In In isometric contraction,the lever arm". There exists a grammar error in this sentence.

"tilting" replaced by "tilt".

6. There are quite a few grammar errors in the paper. Please carefully correct them.

We checked the text throughout and edited when necessary.

7. It will be good if the description of the theoretical part gets clearer.

We have revised the description of the theoretical part, by eliminating unnecessary technical elements. A punctuated outline of the few relevant steps of the analysis is now provided in the main text. We hope that this contributes to making the description clearer. The interested reader can find the details of the derivation in the Methods and in the annexed Supplementary Information.

Reply to Reviewer #3

The authors present a combined experimental-theoretical study of force generation by fast (from psoas muscle) versus slow (from soleus muscle) muscle myosin II isoforms. The myosin II molecules are absorbed onto a micropipette and attached to an actin filament held in an optical trap, which allows to measure force (setup introduced earlier in Ref. 23). A calibration experiments yields that $N=16$ motor heads are building up this "nanomachine". Each of these motors is now assumed to be in one of three states (dissociated, weakly attached or post powerstroke). The corresponding master equation is attacked in many different ways, including moment (or mean field) equations, an adiabatic approximation and stochastic simulations with the Gillespie algorithm. Fitting to the experimental results leads to the main results of this study, namely the parameters reported in Table 1. The fast isoform generates a larger force f_0 , has a smaller duty ratio r and a larger dissociation rate ϕ . Because this work presents results for both fast and slow isoforms, it is an advance over Ref. 23, which was only for the fast version.

The reviewer here does not take into due account other more fundamental advancements: (i) the first is methodological, consisting in recording the nanomachine output in length clamp so that the condition of a very low compliance in series with the motor array, specific (characteristic peculiar) of sarcomere level mechanical experiment in fibers, is recovered; (ii) the second is conceptual, only possible as a consequence of the first, consisting in the re-establishment of the *in situ* condition, that is the motors in the array work in parallel as in the native half-sarcomere and therefore the transition to the steady state force and the force fluctuations depend only on the mechanokinetic properties of the motor isoform; (iii) the third is the consequent, unprecedented possibility to build a self-consistent model to extract the implicit mechanokinetic parameters.

The results are also in good agreement with work from the Yanagida's group (Ref. 22, BPJ 1996). This work is interesting, novel and solid, but highly specialized. It is not clear if it of large interest for the general reader of this journal; it might be more suited for a journal on muscle physiology. To make it more accessible, it should be rewritten such that one can follow it more easily. Technical details should go into the supplement and the text should focus on the essential message. For example, it does not make sense to use equations in the main text that appear only later in the methods section. Either these equations are in the main text, too, or the main text can do without them.

Following the referee remarks, we have removed from the main text technical references to the adiabatic elimination procedure. Also, in the revised version of the paper we avoid explicitly referring in the main text to equations that appear later on in the paper. Also, we now provide a punctuated list to guide the reader

through the main steps of the theoretical analysis. We hope that this helped to improve the clarity and accessibility of the presentation.

Similar with the “nanomachine”: it is introduced rather late and without much emphasis, as if the reader had to know it. When it is finally explained in Fig. 2, then in a very technical manner, with a focus on the control structure. Given its importance (and also the title of the manuscript), it should be introduced rather at the very beginning and with clear explanations regarding its nature and significance.

The Introduction is used first to give the background of the rationale of the experiment and then to demonstrate the unicity of the new methodological approach (the length clamp) to define the performance of the two myosin isoform ensembles in truly isometric condition. Two earlier papers have been dedicated to explaining the power and the limits of the nanomachine design. This is why in this paper the basic design of the nanomachine has been postponed to the point that requires the methodological advancement of eliminating the large trap compliance and recovering the condition for motors to work in parallel. Following the criticisms of Reviewers 1 and 3, this is made now clearer in the Introduction. We hope that with this revised version of the text we have successfully addressed the referee’s concern.

Apart from these concerns regarding general interest and accessibility, I also have a few more scientific concerns. The name “nanomachine” suggests a well-defined setup, but if I understand correctly, the number of motors is determined by an adsorption process at a certain concentration. Why can the authors assume that always around 16 heads are active?

The number of available motors is not an assumption but for each experiment and each functionalized micropipette derives from the recording of the number of rupture events in rigor. The concentration of HMM is set by the titration curve (see ref. 23).

Also in the Supplementary Information we elaborate from a theoretical implication on the implication on varying the imposed system size N.

What makes their setup so deterministic? Why should these 16 heads be all equivalent? I can imagine that interactions with the actin would be very variable along the filament.

That the setup is deterministic was proven in the first paper (ref. 23) by the demonstration that following large sliding (several hundreds of nanometers) in the shortening or lengthening direction the steady isometric force, recovered once the isometric condition is reestablished, is identical to that exhibited by the ensemble at the start of the same interaction. This condition is again demonstrated here. The reason stands in the length of the actin filament being much larger than that of the motor array and in the constancy of the number of motors available in the array. This qualifies the unprecedented power of our system as a model able to mimic the half-sarcomere. Actually, in this respect the synthetic machine is even more efficient than the half-sarcomere, because the amount of sliding for which the number of available motors remains constant is in the millimeter scale, one order of magnitude larger than *in situ* (~100 nm). Thus, the opinion of the reviewer that the interactions would vary along the filament is wrong and generates a series of wrong assertions distributed in the following questions.

*I could imagine other setups, e.g. with DNAorigami, which are much more controlled (compare e.g. Derr, Nathan D., et al. "Tug-of-war in motor protein ensembles revealed with a programmable DNA origami scaffold." *Science* 338.6107 (2012): 662-665).*

The work quoted by the reviewer is at the frontier of technology but so far it does not help to define the emergent properties of an ensemble of non-processive muscle myosin II motors working, at physiological concentration of ATP, as independent force generators. We are convinced, anyway, that a muscle myosin II

array assembled with DNA origami working with 2 mM ATP is an exciting challenging idea for future development of our nanomachine.

For me this setup appears close to the traditional three-bead setup from Ref. 21. Also, there are many other works of this kind with motor ensembles not cited here, e.g. Debold, Edward P., et al. "Direct observation of phosphate inhibiting the force-generating capacity of a mini-ensemble of myosin molecules." Biophysical journal 105.10 (2013): 2374-2384.

The reviewer ignores that the limits of the works using conventional single laser trap systems like TBA for an ensemble of motors to reproduce the properties of the half-sarcomere have been extensively discussed in ref. 23. A striking limit of previous assays is the failure of reproducing the conditions for steady state force generation under physiological ATP concentration. For clarity some relevant arguments discussed in ref. 23 are reported here following:

"In theory the best way to obtain an oriented ensemble of myosin motors consists in using either a native isolated thick filament or a synthetic cofilament made by rods and myosin molecules. With this motor design and the laser trap Kaya and collaborators obtained, in ATP-free solution, static stiffness measurements in agreement with fibre measurements (Piazzesi et al. Cell 131, 784–795 (2007); Kaya & Higuchi, Science 329, 686–689 (2010); Linari et al., J. Physiol. 92, 2476–2490 (2007)). However, the active performance of their motor system (in 1 mM ATP) (Kaya et al. Nat. Commun. 8, 16036 (2017)) consisted only in transient displacements of the actin filament abruptly interrupted after variable extent, without any production of steady force and shortening. Other approaches that exploited the Three Bead Assay geometry (originally designed for single molecule mechanic) by increasing the myosin density on the surface of the fixed bead to have multiple motor interactions (Debold et al., Biophys. J. 89, L34–L36 (2005); Kad et al., Proc. Natl Acad. Sci. USA 102, 16990–16995 (2005)) did not achieve any physiological machine performance, as the [ATP] was systematically kept at least ten times smaller than the millimolar concentration to increase the lifetime of the interactions. The design of our machine, which exploits the dual laser beam technique, appears a decisive choice for the success of the assay. Beyond the large dynamic range, the DLOT geometry ensures that the array of interacting motors lies on a plane that is parallel to the Bead Tailed Actin, in this way preserving their condition of "in parallel force generators". The lack of alignment between the actin filament and the motor ensemble in the single laser trap and in any other existing system based on single actin-single myosin filament (Kaya et al., Nat. Commun. 8, 16036 (2017); 14. Kalganov et al. Biochim. Biophys. Acta 1830, 2710–2719 (2013); Plaçais et al., Phys. Rev. Lett. 103, 158102 (2009)) is likely the reason of their limited performance, as the attached head closest to the bead experiences an additional stress generated by the out-of-axis vertical component of the ensemble force."

A second major issue is that the rates of the model seem not to depend explicitly on ATP-concentration and force. Should k_3 not be linear in ATP-concentration?

The challenging question of this work is to explain the molecular basis of the different mechanical and energetic features of the isometric force of slow and fast muscles taking as a model the synthetic nanomachine working under physiological conditions and released from the confounding contribution of the other sarcomeric proteins. Reducing the ATP concentration in an ensemble of non-processive myosin II motors removes the condition of independent force generators and cannot be in the scope of this work.

And k_1 and k_2 mechanosensitive? Why do the authors not make use of this opportunity to fit to experimental data?

Simply because as far as the experiments in this paper the model has to fit the isometric (length clamped) responses.

I would have expected systematic variation of ATP-concentration. Can this be added?

The manipulation of ATP concentration is a useful tool to increase the time of attachment and reveal the events in single molecule mechanics, but for an array of non-processive muscle myosin II motors working in

parallel the reduction of ATP below the physiological concentration shatters the condition of independent force generators and the underlying kinetic properties, which is not in the aims of this work.

I am missing load sharing, which adds a strongly non-linear aspect to the system and can lead to rupture cascades, as actually observed here.

There is no sign of rupture cascades in our isometric contractions, especially at the steady state of force. This is expected because the isometric condition is that for which the duty ratio of the motors is maximum and thus the probability of interruption of continuous interactions is minimum. In fact the duty ratio reduces with the reduction of the load and the increase of shortening velocity. The observation on the force trace of Fig. 4 by the reviewer indicates that he/she misinterprets the record, missing that the drop in force is caused by the imposition of fast shortening.

As an example for a very similar model that implemented these features, I mention the parallel cluster model by Erdmann, Thorsten, and Ulrich S. Schwarz. "Stochastic force generation by small ensembles of myosin II motors." Physical review letters 108.18 (2012): 188101. Why can the authors here neglect that force has to be shared between the different motor heads?

The kinetics of myosin II of striated muscle evolved to ensure the condition of independent force generator in the isometric performance of the half-sarcomere under physiological ATP concentration. A model must be optimized to simulate this property and not the presence of cascade ruptures that at physiological ATP, as demonstrated above, are absent or, when present, are due to the limits of the motor assay. We hope that our explanations and the related clarifications in the revised text were efficient to clarify the referee's concerns present in his/her second major issue.

Reviewers' comments:

Reviewer #3 (Remarks to the Author):

I thank the authors for their detailed and helpful explanations. I now understand better that a major advance of this paper is the implementation of a length clamp, which rationalizes some of the assumptions of the theory, in particular the notion that the motors work independently like in a half-sarcomere. It is good to see that the authors have now changed the text such that also the general reader will be able to pick up this important aspect of the work. In general, the revised version of this work is much clearer in many aspects and I think that this work should be published in *Communications Biology*. However, I also feel that not all of my concerns have been addressed in the revision, and that the general reader is going to have the same questions when reading the text as I did, so this should be addressed at least as explanations in the text. In the following, I list the questions again from my first report, that I would have liked to be addressed not only in the reply, but also in the main text. This can be easily done in a minor revision.

Fig. 3, determination of number of motors: I maintain that in principle rupture cascades are possible in this experiment. As long as the authors only want to count number of motors, this assay is fine. I now understand that this part of the experiment is not simulated here, but in principle, after each crossbridge dissociation the force balance changes and depending on configuration might make it easier for other crossbridges to rupture. I think this should be explained in the text. Saying that there were no rupture cascades would be correct only if one had made a statistical test on the data, I do not think that this can be decided by eye.

It should be explained better why the same number of motors can be assumed after this rupture experiments. Could it not happen that other motors not bound before now come into play? After all they have been adsorbed from solution onto the pipette (as confirmed by the authors when writing about titration experiments) and as the authors explain later themselves, their orientation is not controlled, so the same should be true for the numbers. When the rupture experiment is repeated two times, does one always get the same number? In my view, this publication should stand by itself and if this has been shown before (Ref. 23 cited by the authors several times when answering my questions), it should be mentioned here. I have no problem with a statistical evaluation of data, but the authors should clearly explain how controlled the number of motors is. Maybe there is an aspect here that I overlook and which makes this more deterministic than I understand now.

ATP-concentration: I understand that the authors are not interested in changing this and always keep the physiological value, but I think that having this nice nanomachine, it is an interesting option and this could be mentioned e.g. for future perspectives. Also I suggest to at least mention how ATP-concentration would go into the rates of the model, in my view this is an essential part of any crossbridge cycle model, even if the authors decide not to exploit it here.

Last comment: Table 1: this is the central result of this work, but the three parameters (motor force f_0 , duty ratio r , flux ϕ) are not defined there. I think this should be in the caption. Same for Fig. 5, where the parameters should be explained in the caption.

Reply to Reviewer #3 (reviewer's comments in blue)

I thank the authors for their detailed and helpful explanations. I now understand better that a major advance of this paper is the implementation of a length clamp, which rationalizes some of the assumptions of the theory, in particular the notion that the motors work independently like in a half-sarcomere. It is good to see that the authors have now changed the text such that also the general reader will be able to pick up this important aspect of the work. In general, the revised version of this work is much clearer in many aspects and I think that this work should be published in Communications Biology. However, I also feel that not all of my concerns have been addressed in the revision, and that the general reader is going to have the same questions when reading the text as I did, so this should be addressed at least as explanations in the text. In the following, I list the questions again from my first report, that I would have liked to be addressed not only in the reply, but also in the main text. This can be easily done in a minor revision.

Fig. 3, determination of number of motors: I maintain that in principle rupture cascades are possible in this experiment. As long as the authors only want to count number of motors, this assay is fine. I now understand that this part of the experiment is not simulated here, but in principle, after each crossbridge dissociation the force balance changes and depending on configuration might make it easier for other crossbridges to rupture. I think this should be explained in the text.

The pull on the array in rigor is exerted at an angle with the x-axis. This allows a higher force to be exerted on the first attached motor (starting from the bead side), while the others, almost aligned on the x-axis, share only the force component along the x-axis. In this way: (i) acto-myosin bonds rupture one at a time and (ii) a motor cannot reattach once it is detached. After each detachment the force drops because the length of actin filament segment between the bead and the next attached motor is increased. Thus an additional pulling is necessary to get to the next rupture event, the occurrence of which will vary according to the distance between the two neighboring motors. A breaking cascade is in principle prevented by the fact that after each rupture the overall force drops by the lengthening of the actin segment between the bead and the next motor. We have taken into account the reviewer question and given the details to better understand the rupture protocol.

Saying that there were no rupture cascades would be correct only if one had made a statistical test on the data, I do not think that this can be decided by eye.

As you can see from the Fig. 1 here, the absence of rupture cascades relies also on the frequency distribution of the number of events per interaction. Data are collected with different bead tailed actin filaments on either the same micropipette or different micropipettes with similar diameter ($3.5 \pm 0.5 \mu\text{m}$).

It should be explained better why the same number of motors can be assumed after this rupture experiments. Could it not happen that other motors not bound before now come into play? After all they have been adsorbed from solution onto the pipette (as confirmed by the authors when writing about titration experiments) and as the authors explain later themselves, their orientation is not controlled, so the same should be true for the numbers. When the rupture experiment is repeated two times, does one always get the same number?

Within the limits shown by the gaussian frequency distribution.

In my view, this publication should stand by itself and if this has been shown before (Ref. 23 cited by the authors several times when answering my questions), it should be mentioned here. I have no problem with a statistical evaluation of data, but the authors should clearly explain how controlled the number of motors is. Maybe there is an aspect here that I overlook and which makes this more deterministic than I understand now.

We have integrated the text to make clearer the degree of reliability of the method.

ATP-concentration: I understand that the authors are not interested in changing this and always keep the physiological value, but I think that having this nice nanomachine, it is an interesting option and this could be mentioned e.g. for future perspectives. Also I suggest to at least mention how ATP-concentration would go into the rates of the model, in my view this is an essential part of any crossbridge cycle model, even if the authors decide not to exploit it here.

Our longstanding experience with sarcomere-level mechanics on isolated muscle fibres, either intact or skinned, suggests that reducing the ATP concentration below the physiological level on an ensemble of motors working on the same actin filament opens up a scenario in which the effect of said reduction on the rates of the single actin-myosin interaction cycle is complicated by the addition of the effects due to the presence of the motors attached as first that have already achieved the end of the chemomechanical cycle and are waiting for an ATP to bind to be able to detach: these motors would resist further sliding promoted by the new motors and thus also in isometric conditions oppose/slow down the increase of strain of the elastic element in series to the array (remaining in a limited extent also in our LC conditions) and thus of force. In this respect the titration of the effect of ATP concentration is a convenient protocol only for investigation of single molecule kinetics. To extend our investigation to the consequences of the reduction of ATP on the performance of a motor ensemble is not the aim of this work and, to be taken into consideration, would require a LC control even more efficient than the one achieved for these experiments.

Last comment: Table 1: this is the central result of this work, but the three parameters (motor force f_0 , duty ratio r , flux ϕ) are not defined there. I think this should be in the caption. Same for Fig. 5, where the parameters should be explained in the caption.

Required definition of parameters added in both cases.

Fig. 1. Frequency distribution of number of rupture events per rigor interaction of the psoas HMM ensemble. Data are fitted with a Gaussian with center = 7.8 and standard deviation $\sigma = 1.3$ ($n = 26$).